# Approximate Euclidean lengths and distances beyond Johnson-Lindenstrauss

**Aleksandros Sobczyk**
IBM Research and ETH Zürich
Zürich, Switzerland
obc@zurich.ibm.com

**Mathieu Luisier**
ETH Zürich
Zürich, Switzerland
mluisier@iis.ee.ethz.ch

## Abstract

A classical result of Johnson and Lindenstrauss states that a set of $n$ high dimensional data points can be projected down to $O(\log n/\epsilon^2)$ dimensions such that the square of their pairwise distances is preserved up to a small distortion $\epsilon \in (0, 1)$. It has been proved that the JL lemma is optimal for the general case, therefore, improvements can only be explored for special cases. This work aims to improve the $\epsilon^{-2}$ dependency based on techniques inspired by the Hutch++ Algorithm [34], which reduces $\epsilon^{-2}$ to $\epsilon^{-1}$ for the related problem of implicit matrix trace estimation. We first present an algorithm to estimate the Euclidean lengths of the rows of a matrix. We prove for it element-wise probabilistic bounds that are at least as good as standard JL approximations in the worst-case, but are asymptotically better for matrices with decaying spectrum. Moreover, for any matrix, regardless of its spectrum, the algorithm achieves $\epsilon$-accuracy for the total, Frobenius norm-wise relative error using only $O(\epsilon^{-1})$ queries. This is a quadratic improvement over the norm-wise error of standard JL approximations. We also show how these results can be extended to estimate (i) the Euclidean distances between data points and (ii) the statistical leverage scores of tall-and-skinny data matrices, which are ubiquitous for many applications, with analogous theoretical improvements. Proof-of-concept numerical experiments are presented to validate the theoretical analysis.

## 1 Introduction

The Johnson-Lindenstrauss (JL) lemma [29] is a fundamental concept in dimensionality reduction and data science. Given a set of $n$ high dimensional data points $X = \{x_1, ..., x_n\}$, where each $x_i \in \mathbb{R}^d$, the goal is to find a projection $f : \mathbb{R}^d \to \mathbb{R}^k$ that maps the vectors to a much smaller dimension $k \ll d$ such that the geometry of the original set is approximately preserved. Specifically, the projection should preserve the pairwise distances up to a small distortion $\epsilon \in (0, 1)$, that is

$$(1 - \epsilon)\|x_i - x_j\|^2 \leq \|f(x_i) - f(x_j)\|^2 \leq (1 + \epsilon)\|x_i - x_j\|^2, \tag{1}$$

for all $i, j \in [n]$. If $f$ satisfies this property, then it is called an $\epsilon$-isometry. Johnson and Lindenstrauss proved that, given $\epsilon$, such an $f$ can be found in randomized polynomial time and that the projected dimension is no larger than $O(\log n/\epsilon^2)$. In the last decades the JL lemma has made an impact in many areas, including Graph Algorithms [6, 42], Machine Learning [5, 11, 16, 24], Numerical Linear Algebra [14, 33, 40, 44] and Optimization [20, 22, 38].

In the existing literature, a common approach to approximate the metric is to first find a map that preserves Euclidean lengths instead of distances. The approximate isometry property is then achieved by applying this map to all the pairwise difference vectors, since the Euclidean distance between $x$ and $y$ is equal to the length of $x - y$; cf. [18, 29]. In this work we follow the same methodology. We first study the problem of approximating the Euclidean lengths of the rows of an

arbitrary matrix $A \in \mathbb{R}^{n \times d}$ in the so-called matrix-vector query model. In this model, the matrix $A$ might not be explicitly available, but we have access to a linear operator that computes the product $Ax$, for an arbitrary vector $x$. For non-symmetric and rectangular matrices we assume that we can compute both $Ax$ and $A^\top x$. This model is particularly useful when dealing with matrix functions, i.e. when $A = f(B)$ is implicitly defined as a function of another matrix $B$. Two applications arise from network science. For a graph $G$, let $A$ be its adjacency matrix, $B$ its edge-incidence matrix, $W$ the diagonal matrix containing non-negative edge weights and $L = B^\top W B$ its Laplacian. The exponential function of the adjacency matrix, $e^{\beta A}$, provides information about node centrality measures [10, 23], while the diagonal entries of $BL^\dagger B$ are the so-called "effective resistances" of the edges [33, 36, 42], which can be used to identify important edges. Another example, which is detailed in Section 4, is for computing the leverage scores of a matrix, which can be found in the diagonal of the orthogonal projector $H = A(A^\top A)^\dagger A^\top$. In all of these applications, explicitly evaluating the corresponding matrix functions has typically cubic complexity, which can be prohibitively expensive. However, after some algebra, the quantities of interest can be expressed as the Euclidean lengths of the rows of a matrix function in the matrix-vector query model. One can therefore use techniques related to the JL lemma to derive fast approximations. To this end, we first recall the concept of Johnson-Lindenstrauss transforms, as defined in [40].

**Definition 1** (Johnson-Lindenstrauss transform [40])**.** *A random matrix $S \in \mathbb{R}^{r \times d}$ forms a Johnson-Lindenstrauss transform with parameters $\epsilon, \delta \in (0, 1/2)$ and positive integer $n$, or $(\epsilon, \delta, n)$-JLT for short, if with probability at least $1 - \delta$, for any fixed set $V \subseteq \mathbb{R}^d$ with $n$ elements it holds that $(1 - \epsilon)\|v\|^2 \leq \|Sv\|^2 \leq (1 + \epsilon)\|v\|^2$ for all $v \in V$.*

It is known that Gaussian matrices can provide JLTs; c.f. [5, 29].

**Lemma 1** (Gaussian random projections [5, 29])**.** *Let $G \in \mathbb{R}^{r \times d}$ with i.i.d. elements from $\mathcal{N}(0, 1/\sqrt{r})$ and $\epsilon \in (0, 1/2)$. For a fixed $x \in \mathbb{R}^d$ it holds that*

$$\Pr\left[\left|\|x\|^2 - \|Gx\|^2\right| \leq \epsilon\|x\|^2\right] \geq 1 - 2\exp\left(-\frac{r(\epsilon^2 - \epsilon^3)}{4}\right).$$

*For a set $X \subset \mathbb{R}^d$ of $n$ vectors and for $\delta \in (0, 1/2)$, as long as $r \geq \frac{4\log(2n/\delta)}{\epsilon^2 - \epsilon^3}$, then $G$ forms an $(\epsilon, \delta, n)$-JLT.*

The dimension $r$ of $G$ depends on $1/\epsilon^2$, which can quickly become very large if a high accuracy is needed. Consequently, if $r$ is very large, then it is also very expensive to compute the product $GA$. There is therefore no advantage in taking an approximate solution over computing the true solution. Here, we would like to improve this dependency on $\epsilon$. To achieve this, we will also use a powerful generalization of JLT, the so-called of Oblivious Subspace Embeddings [40], which extend the JLT definition for an entire subspace, instead of a finite set. We use the definitions from [44].

**Definition 2** (Oblivious Subspace Embedding [44])**.** *Let $\mathcal{D}$ be a distribution on $r \times n$ matrices $S$, where $r$ is a function of $n, d$ and $\epsilon, \delta \in (0, 1/2)$. We call $S$ an $(\epsilon, \delta)$ Oblivious Subspace Embedding, or $(\epsilon, \delta)$-OSE if for any fixed $n \times d$ matrix $A$, $S \sim \mathcal{D}$ is a $(1 \pm \epsilon)$ $l_2$-Subspace Embedding for $A$ with probability at least $1 - \delta$, that is, for all $x \in \mathbb{R}^d$ it holds that*

$$(1 - \epsilon)\|Ax\|^2 \leq \|SAx\|^2 \leq (1 + \epsilon)\|Ax\|^2.$$

**Notation.** By default, the Householder notation is used, denoting matrices with capital letters, vectors with small letters, and scalars with Greek letters. $[n]$ is the set $\{1, 2, ..., n\}$, where $n \in \mathbb{N}$. $I_n$ is the identity matrix of size $n \times n$ and $e_i$ its $i$-th column. $A_{i,j}$ is the element of $A$ in row $i$ and column $j$. $A_k$ denotes the best rank-$k$ approximation of $A$ in the 2-norm. $\|A\|_F$ is the Frobenius norm of $A$, while the 2-norm is assumed for both matrices and vectors when the norm subscript is omitted. $A^\top$ is the transpose of $A$ and $A^\dagger$ is the pseudoinverse. $\mathbb{P}[\alpha] \in [0, 1]$ denotes the probability of an event $\alpha$ to occur. $\mathcal{N}(\mu, \sigma)$ is the normal distribution with mean value $\mu$ and standard deviation $\sigma$. $\sigma_i(A)$ denotes the $i$-th largest singular value of $A$. $\texttt{nnz}(A)$ is the number of nonzeros $A$. $\tilde{O}(k) := O(k \log^c(k))$ for some constant $c$. We refer to matrices with i.i.d. elements from $\mathcal{N}(0, 1)$ as Gaussian matrices. In the complexity analysis, $\omega$ denotes the fast matrix multiplication exponent, where $2 \leq \omega < 2.37286$ [4].

**Why Gaussians?** In this work we focus on Gaussian random projections. Other constructions satisfying the $(\epsilon, \delta, n)$-JLT definition exist in the literature, such as randomized Fourier/Hadamard [2, 3, 43] or sparse [1, 15, 17, 30, 32, 35] transforms. When the input matrix is explicitly available,

such constructions are faster to apply than Gaussian random projections. However, as already mentioned, this is not the case in the matrix-vector query model. Consider the following simple example. Assume that we are interested to compute the Euclidean row norms of the matrix $A^2$, where $A \in \mathbb{R}^{n \times n}$ is a dense input matrix. As already detailed, these norms can be approximated by the Euclidean row norms of the matrix $\tilde{A} = A^2 S$, where $S$ is an $(\epsilon, \delta, n)$-JLT matrix with $r$ columns. If $S$ is a Gaussian matrix, then $\tilde{A}$ can be evaluated in two steps, i.e. by computing $B = AS$ and then $\tilde{A} = AB$, with a total complexity of $O(n^2 r^{\omega-2})$. On the other hand, if $S$ is a CountSketch [15, 35], then the matrix $B = AS$ can be evaluated in $O(n^2)$, which is faster than $O(n^2 r^{\omega-2})$. However, since $B$ is dense and it has no special structure, then the second step $\tilde{A} = AB$ still requires $O(n^2 r^{\omega-2})$ operations. The total complexity is therefore still dominated by the $O(n^2 r^{\omega-2})$ term. It is also known that Gaussian matrices require asymptotically less columns $r$ than the aforementioned fast transforms to satisfy the $(\epsilon, \delta, n)$-JLT definition. This means that Gaussian matrices can in fact be the fastest option, since $r$ is minimized. Nevertheless, all results of this work are derived as structural results, i.e. they do not necessarily require Gaussian matrices. Any matrix satisfying the properties that are detailed in the proofs can be used instead. All aforementioned fast transforms are for example excellent candidates.

**Contributions.** In Algorithm 1, we present the main algorithm of this work to approximate the Euclidean row norms of a matrix $A \in \mathbb{R}^{n \times d}$, which is inspired by the Hutch++ algorithm [34]. Following [34], this algorithm is called "Adaptive," since it needs to make two passes over the input matrix $A$. The main contributions are the following:

1. The proposed algorithms require asymptotically less matrix-vector queries to achieve the same accuracy as standard JL random projections for matrices with decaying spectrum, that is, spectral decay properties are reflected in the approximation bounds. To the best of our knowledge, this is the first work to provably reduce the number of required matrix-vector queries for Euclidean length approximations.

2. For any matrix, regardless of its spectrum, the proposed algorithms require a number of matrix-vector queries that depends on $1/\epsilon$ to achieve $\epsilon$-accuracy for the total, Frobenius norm-wise error, as opposed to $1/\epsilon^2$ for standard JL.

3. For the worst-case inputs, that is, for matrices with flat spectrum, the approximated values are at least as good as standard JL.

4. The techniques can be directly applied to and give similar improvements for the related problems of approximate pairwise Euclidean distances and approximate leverage scores.

---

**Algorithm 1** Adaptive Euclidean Norm Estimation

---

**Input:** Matrix $A \in \mathbb{R}^{n \times d}$, $n \geq d$, positive integer $m < d$.
**Output:** $\tilde{x}_i \approx \|e_i^\top A\|^2$.
  # Step 1: Low-rank approximation
  1: Construct two random matrices $S, G \in \mathbb{R}^{d \times m}$ with i.i.d. elements from $\mathcal{N}(0, 1)$.       $\triangleright O(dm)$
  2: Compute $B = A^\top (AS)$.                                                                $\triangleright O(T_{\mathrm{MM}}(A, m))$
  3: Compute an orthonormal basis $Q \in \mathbb{R}^{d \times m}$ for $\mathrm{range}(B)$ (e.g., via QR).             $\triangleright O(dm^2)$
  # Step 2: Project and compute row norms
  4: Compute $\tilde{A} = AQ$ and $C = AG$.                                                      $\triangleright O(T_{\mathrm{MM}}(A, m))$
  5: Compute $\tilde{\Delta} = A(I - QQ^\top)G = C - \tilde{A}(Q^\top G)$.                                   $\triangleright O(nm^2)$
  6: **return** $\tilde{x}_i = \|e_i^\top \tilde{A}\|^2 + \|e_i^\top \tilde{\Delta}\|^2$, for all $i \in [n]$.                        $\triangleright O(nm)$

---

In Table 1 we summarize the approximation guarantees of the proposed Algorithms 1, 2, and 3 for the aforementioned problems. We also compare it to the corresponding bounds of existing JL-based approximations to highlight the achieved improvements. For the precise statements we refer to the corresponding sections.

Regarding the complexity of Algorithm 1, by $T_{\mathrm{MM}}(A, m)$ we denote the complexity of computing the product $AB$, where $B$ is a dense matrix with $m$ columns. For example, if $A$ is just a dense matrix, $T_{\mathrm{MM}}(A, m) = O(ndm^{\omega-2})$, by leveraging fast matrix multiplication [4]. Another example is when $A$ is implicitly available as a function of a given sparse matrix $C$, e.g. if $A = C^3$ then

Table 1: Comparison between the approximation bounds that are achieved in this work versus standard JL random projections for the three different problems considered here. In all cases, the number of matrix-vector queries $m$ that are performed is the same. It is proportional to $\epsilon^{-2}$ (up to logarithmic factors on $n, \delta$), where $\epsilon \in (0, 1/2)$ is the accuracy and $\delta \in (0, 1/2)$ the success probability. Here, $k$ is an integer such that $m = \Omega(k/\delta)$ and $\bar{A}_k = A - A_k$. $M$ is a matrix such that its rows define pairwise distance vectors between the rows of $A$. The $\theta_i$'s are the leverage scores of the input matrix.

| | Element-wise | | Frobenius norm-wise | | |
| --- | --- | --- | --- | --- | --- |
| | This work | JL | This work | JL | ref. |
| Row norms | $\epsilon \\|e_i^\top A\\| \\|e_i^\top \bar{A}_k\\|$ | $\epsilon\\|e_i^\top A\\|^2$ | $\epsilon^2\\|A\\|_F^2$ | $\epsilon\\|A\\|_F^2$ | Thms. 1 & 2 |
| Distances | $\epsilon \\|e_i^\top M\\| \\|e_i^\top \bar{M}_k\\|$ | $\epsilon\\|e_i^\top M\\|^2$ | $\epsilon^2\\|M\\|_F^2$ | $\epsilon\\|M\\|_F^2$ | Thm. 5 |
| Leverage scores | $\epsilon\theta_i$ | $\epsilon\theta_i$ | $\epsilon^2 d$ | $\epsilon d$ | Thm. 3 |

$T_{\mathrm{MM}}(A, m) = 3 \times T_{\mathrm{MM}}(C, m) = O(\texttt{nnz}(C)m)$. The results are stated for general $T_{\mathrm{MM}}(A, m)$, but they will be specialized, where applicable, for the targeted applications.

**Related work.** A related topic is stochastic matrix trace estimation [7, 27, 28, 34, 37, 39]. Intuitively, a set of data points can be seen as the columns of a matrix. In various applications the trace of such a matrix contains useful information like triangle counts in graphs [6]. Hutchinson [27] proposed a randomized algorithm to rapidly approximate the trace of such a matrix, which uses similar ideas to JL: it projects the rows of the matrix onto a low-dimensional subspace so that the trace can be quickly computed. Avron and Toledo showed that the dimension of that subspace needs to be proportional to $\epsilon^{-2}$ in order to guarantee a worst-case $\epsilon$-approximation for the trace [7]. This dependence on $\epsilon$ matches the requirements for the $\epsilon$-isometry of JL. The $\epsilon^{-2}$ overhead can be prohibitive when $\epsilon$ is small, i.e. in applications where high accuracy is needed. Recently, in their seminal work, Meyer, Musco, Musco, and Woodruff [34] proved a remarkable result: their Hutch++ algorithm is the first to obtain $\epsilon$-accuracy for stochastic trace estimation while requiring only $1/\epsilon$ matrix-vector queries. For the related problem of estimating the diagonal elements of a matrix, which was also recently studied in depth [9, 26], Baston and Nakatsukasa [9] achieved $\epsilon$-accuracy for the total, norm-wise error of the entire diagonal using $O(1/\epsilon)$ matrix-vector queries, but not for each individual diagonal element, which should not be possible due to the optimality of the JL lemma [31]. It is worth noting that the squared row norms of a matrix $A$ can be found in the diagonal of $AA^\top$, therefore, our work is closely related. Our results for the total norm-wise error, however (see e.g. Theorem 2), are tighter than simply using [9] on $AA^\top$, since we are exploiting the special structure of $AA^\top$. From a Fine-Grained complexity perspective, estimating row norms can be easily reduced to diagonal estimation, but the opposite reduction is not straightforward, therefore, one can argue that diagonal estimation is harder, which justifies our tighter bounds.

**Outline.** The analysis of Algorithm 1 is given in Section 2. In Sections 3 and 4 we show two important applications of the main results, namely for the estimation of the pairwise Euclidean distances between a set of data points and for the estimation of the statistical leverage scores of a tall-and-skinny data matrix. In Section 5 we present indicative experiments to validate the theoretical analysis, before finally giving concluding remarks and future directions in Section 6.

## 2 Analysis of Algorithm 1

In this section we provide the analysis of Algorithm 1. Preliminary results and long proofs which were omitted from the main text and can be found in the Appendix. We state the following general result for the element-wise bounds of Algorithm 1.

**Lemma 2.** *(Proof in the Appendix) Let $A \in \mathbb{R}^{n \times d}$. If we use Algorithm 1 with $m$ matrix-vector queries to estimate the Euclidean lengths of the rows $e_i^\top A$, $i \in [n]$, then as long as $m \geq l \geq 32 \log(4n/\delta)$ it holds that*

$$\left| \tilde{x}_i - \\|e_i^\top A\\|^2 \right| \leq \sqrt{\tfrac{8 \log(\frac{2n}{\delta})}{l}} \\|e_i^\top A(I - QQ^\top)\\|^2, \text{ for all } i \in [n],$$

*with probability at least $1 - \delta$ for all $i \in [n]$ simultaneously.*

Evidently, this result implies that if we can determine a suitable bound for $\|e_i^\top A(I - QQ^\top)\|^2$ then we automatically get a proper bound for the element-wise approximations of Algorithm 1. If $A$ has a fast decaying spectrum and $Q$ captures the dominant eigenspace of $A$ we can expect that our approximations are very accurate, even for small $l$. For the general case, however, the following Lemma 3 as well as the optimality of the JL lemma [31] already hint that this is not possible (see also Appendix II, Limitations of low-rank projections).

**Lemma 3.** *Let* $A \in \mathbb{R}^{n \times d}$. *For* $1 \leq k < d$, *it holds that* $\|e_i^\top (A - A_k)\|_2^2 \leq \sigma_{k+1}^2(A) \leq \frac{\|A_k\|_F^2}{k}$.

*Proof.* Clearly, $\|e_i^\top (A - A_k)\|_2^2 \leq \max_{\|x\|=1} \|x^\top (A - A_k)\|_2^2 = \sigma_{k+1}^2(A)$. For the second part we have that $\sigma_{k+1}^2(A) \leq \frac{1}{k} \sum_{i=1}^{k} \sigma_i^2(A) = \frac{\|A_k\|_F^2}{k}$. □

## 2.1 Projecting rows on randomly chosen subspaces

To proceed further with the analysis, we show some length-preserving properties of the orthogonal projector $QQ^\top$, which is an orthogonal projector on a random subspace as obtained in line 3 of Algorithm 1. Note that Corollary 1 is stated for constant factor approximations. Here we provide a brief proof sketch. For the main result we refer to Lemma 8 in Appendix III.

**Corollary 1** (Projection on $\mathrm{rowspace}(SA^\top A)$). *(Proof in the Appendix) Let* $\delta \in (0, \frac{1}{2})$, $\bar{A}_k = A - A_k$, *and $S$ be such that*

(i) $S \sim \mathcal{D}$, *where $\mathcal{D}$ is an $(1/3, \delta)$-OSE for any fixed $k$-dimensional subspace;*

(ii) $S$ *is a* $(1/3, \delta, 2n)$-JLT.

*If $Q$ is a matrix that forms an orthonormal basis for* $\mathrm{rowspace}(SA^\top A)$, *then, with probability at least $1 - 2\delta$, for all $i \in [n]$ simultaneously, it holds that*

$$\|e_i^\top A(I - QQ^\top)\|^2 \leq \|e_i^\top (\bar{A}_k)\|^2 + \frac{1}{2} \frac{\sigma_{k+1}^2(A)}{\sigma_k^2(A)} \|e_i^\top A_k\| \|e_i^\top \bar{A}_k\| \leq \frac{3}{2} \|e_i^\top A\| \|e_i^\top \bar{A}_k\|.$$

*Proof sketch.* To prove the result it suffices to find a projector within $\mathrm{rowspace}(SA^\top A)$ with the desired properties. To do this, we consider the matrix $\Pi_k = V_k(SV_k\Sigma_k^2)^\dagger SA^\top A$, where $V_k, \Sigma_k$ originate from the SVD of $A_k = U_k\Sigma_k V_k^\top$. Clearly, this $\Pi_k$ is a rank-$k$ matrix within $\mathrm{rowspace}(SA^\top A)$. After some algebra, the problem reduces to get a bound for the quantities $|e_i^\top A V_k \Sigma_k^2 C^{-1} V_k^\top S^\top S \bar{V}_k \bar{\Sigma}_k^2 \bar{V}_k^\top A^\top e_i|$, for all $i \in [n]$, where the existence of $C^{-1}$ is guaranteed due to the $(1/3, \delta)$-OSE property of $S$. For each $i$, this quantity is the absolute value of the inner product $\langle S(V_k C^{-1} V_k^\top A^\top)e_i, S(\bar{V}_k \bar{V}_k^\top A^\top)e_i \rangle$, which can be written in a simplified form as $\langle Sx_k, S\bar{x}_k \rangle$. Therefore, we use an $(1/3, \delta, 2n)$-JLT to bound the inner products between the vectors of the set

$$V = \left\{ e_i^\top A V_k \Sigma_k^2 C^{-1} V_k^\top | i \in [n] \right\} \bigcup \left\{ e_i^\top A \bar{V}_k \bar{\Sigma}_k^2 \bar{V}_k^\top | i \in [n] \right\}.$$

□

Having all pieces in-place, we can finally bound the element-wise approximations of Algorithm 1.

**Theorem 1.** *(Proof in the Appendix) Let* $A \in \mathbb{R}^{n \times d}$ *and* $n \geq d$. *If we use Algorithm 1 with $m$ matrix-vector queries to estimate the Euclidean lengths of the rows of $A$, then there exists a global constant $C$ such that, as long as*

(i) $m \geq l \geq O(\log(n/\delta))$, *such that $G$ satisfies Lemma 1 and $S$ forms an $(1/3, \delta, 2n)$-JLT,*

(ii) $m \geq O(k + \log(1/\delta))$, *such that $S$ forms an $(1/3, \delta)$-OSE for a $k$-dimensional subspace,*

*then it holds that*

$$\left| \tilde{x}_i - \|e_i^\top A\|^2 \right| \leq C\sqrt{\frac{\log(\frac{n}{\delta})}{l}} \|e_i^\top (A - A_k)\| \|e_i^\top A\| \leq C\sqrt{\frac{\log(\frac{n}{\delta})}{lk}} \|A_k\|_F \|e_i^\top A\|,$$

*for all $i \in [n]$ with probability at least $1 - 3\delta$.*

**Discussion.** We can investigate the bounds for special matrix cases. We highlight the approximation power of Algorithm 1 for matrices with decaying spectrum. For matrices with a linear decay it suffices to take $m \gtrsim O(\epsilon^{-1}\sqrt{\log(n/\delta)})$ queries to achieve an almost $\epsilon$-accuracy. For matrices with exponential decay we can use as few as $m \geq O(\log(1/\epsilon))$ matrix-vector queries. For matrices with no decay, e.g., for orthogonal projector matrices, Lemma 2 already guarantees that Algorithm 1 provides at least as accurate element-wise approximations as standard JL projections. We recall once more that the JL lemma is optimal in the general case [31], therefore, improvements can only be derived for special cases, like the ones considered here.

## 2.2 Frobenius norm bounds

Due to the tightness of Lemma 3, which is crucial for the element-wise bounds, it is highly unlikely that low-rank projection-based methods can generally achieve better element-wise approximations. However, if we carefully examine the total, Frobenius norm-wise error, we can in fact obtain a true $\epsilon$-relative error approximation. This cannot be done by "simply" adding together all element-wise bounds, i.e., we must use a different "collective" approach. This result also makes the element-wise bounds more appealing: even if there remain few outliers that violate the element-wise $\epsilon$-approximation, the total error is still very small. We note that this is a quadratic improvement over the norm-wise error of standard JL projections.

**Theorem 2.** *(Proof in the Appendix) In Algorithm 1, for some absolute constants $c, C$, if $l > c\log(1/\delta)$, it holds that*

$$\left|\tilde{X} - \|A\|_F^2\right| \leq C\sqrt{\frac{\log(\frac{1}{\delta})}{lk}}\|A\|_F^2,$$

*where $\tilde{X}$ is the sum of the returned approximations. For $l = k = O\left(\frac{\sqrt{\log(\frac{1}{\delta})}}{\epsilon}\right)$, where $\epsilon \in (0, 1/2)$, setting $m \geq O(k/\delta + \log(\frac{1}{\delta}))$, it follows that*

$$\left|\tilde{X} - \|A\|_F^2\right| \leq \epsilon\|A\|_F^2.$$

## 2.3 Complexity

The complexity of Algorithm 1 is as follows. In the first step two matrices $S$ and $G$ must be generated with $d \times m$ random elements each. Hence, $O(dm)$ calls to a random number generator are required. In the second step, the products $A^\top(AS)$ and $A^\top(AG)$ can be both computed in $O(T_{\text{MM}}(A, m) + T_{\text{MM}}(A^\top, m))$. Next we need to create an orthonormal basis for $A^\top AS$ which has size $d \times m$. This can be done with a standard Householder QR or another orthogonal factorization in $O(dm^2)$ [25, Chapter 5]. The complexity of this operation can be improved using fast matrix multiplication primitives [19]. The product $AQ$ costs $O(T_{\text{MM}}(A, m))$. We then have to compute $\tilde{A}(Q^\top G)$, which takes $O(dm^2)$ or $O(dm^{\omega-1})$ to first obtain $Q^\top G$ and then the same cost to get $\tilde{A}(Q^\top G)$. Finally, for the last step the squared row norms of $2n$ vectors, the rows of $\tilde{A}$, and the rows of $\tilde{\Delta}$, are needed. For each row of $\tilde{A}$ and $\tilde{\Delta}$ the cost of computing the squared Euclidean norm is $O(m)$, therefore the cost for the last step is $O(nm)$. Summing up, the total cost of Algorithm 1 is $O(dm^2 + T_{\text{MM}}(A, m) + nm)$.

# 3 Euclidean distances

In many applications it is desired to find an approximate isometry for a set of data points. Let $A \in \mathbb{R}^{n \times d}$ be a matrix whose rows define these $d$-dimensional data points. Assume we are interested to estimate all the $\binom{n}{2}$ distances between the rows of $A$. Let $B$ be a matrix with size $\binom{n}{2} \times n$ and each row of $B$ is equal to the vector $(e_i - e_j)^\top$ for some $i, j \in [n]$.[1] Each row $(e_i - e_j)^\top$ of $B$, when multiplied with $A$, gives the difference vector $e_i^\top A - e_j^\top A$. Therefore, to estimate the Euclidean distances between the rows of $A$, it is sufficient to estimate the Euclidean lengths of the rows of the matrix $BA$. In Algorithm 2 we describe this procedure for a general "incidence matrix" $B$, e.g., when one wants to estimate only a subset of the pairwise distances. Since $B$ has in general more rows than $A$, the matrix multiplications must be computed in the correct order to minimize their complexity.

---

[1] Note that $B$ is nothing more than the edge incidence matrix of a complete graph.

---

**Algorithm 2** Adaptive Euclidean Distance Estimation

---

**Input:** Data matrix $A \in \mathbb{R}^{t \times d}$, $t \geq d$, incidence matrix $B \in \mathbb{R}^{n \times t}$, positive integer $m < d$.
**Output:** Approximate pairwise distances $\tilde{x}_i \approx \|e_i^\top BA\|^2$, $i \in [n]$.
    # Step 1: Low-rank approximation
  1: Construct two random matrices $S, G \in \mathbb{R}^{d \times m}$ with i.i.d. elements from $\mathcal{N}(0,1)$.     $\triangleright O(dm)$
  2: Compute the product $\tilde{S} = A^\top(B^\top(B(AS)))$.     $\triangleright O(T_{\mathrm{MM}}(A, m) + T_{\mathrm{MM}}(A^\top, m) + nm)$
  3: Compute an orthonormal basis $Q \in \mathbb{R}^{d \times m}$ for $\mathrm{range}(\tilde{S})$ (e.g., via QR).     $\triangleright O(dm^2)$
    # Step 2: Project and compute row norms
  4: Compute $\tilde{A} = AQ$ and $C = AG$.     $\triangleright O(T_{\mathrm{MM}}(A, m))$
  5: Compute $\tilde{\Delta} = A(I - QQ^\top)G = C - \tilde{A}(Q^\top G)$.     $\triangleright O(tm^2)$
  6: **return** $\tilde{x}_i = \|(e_i^\top B)\tilde{A}\|^2 + \|(e_i^\top B)\tilde{\Delta}\|^2$, for all $i \in [n]$.     $\triangleright O(nm)$

---

**Bounds.** Approximation bounds can be directly derived from Theorems 1 and 2, replacing $A$ with $BA$. For completeness, they can be found in Theorem 5 in the Appendix.

**Complexity.** The complexity of Algorithm 2 is as follows. $O(dm)$ operations are needed to generate $G$ and $S$. The product $\tilde{S} = A^\top B^\top BAS$ is evaluated in three steps. We first compute $AS$ in $O(T_{\mathrm{MM}}(A, m))$, then $B(AS)$ and $B^\top(BAS)$ in $O(nm)$, and finally $A^\top(B^\top BAS)$ in $O(T_{\mathrm{MM}}(A^\top, m))$. The intermediate products can be calculated in batches to save memory. The QR factorization of $\tilde{S}$ requires $O(dm^2)$. The products $\tilde{A} = AQ$ and $C = AG$ both require $O(T_{\mathrm{MM}}(A, m))$, whereas the product $Q^\top G$ can be performed in $O(dm^2)$. Accordingly, the product $\tilde{A}(Q^\top G)$ needs $O(tm^2)$ and $C - \tilde{A}(Q^\top G)$ $O(tm)$. In the last step each row norm costs $O(m)$ operations, resulting in $O(nm)$.[2] The total complexity of Algorithm 2 is therefore

$$O\left(T_{\mathrm{MM}}(A, m) + T_{\mathrm{MM}}(A^\top, m) + nm + dm^2 + tdm\right).$$

## 4 Statistical leverage scores

We next consider the problem of approximating the leverage scores of a tall-and-skinny matrix. The leverage scores of the rows of $A$ can be found in the diagonal of the orthogonal projector matrix $P = AA^\dagger = UU^\top$, where $U$ is any orthonormal basis for $\mathrm{range}(A)$. Specifically, the leverage score $\theta_i$ of the $i$-th row of $A$ is equal to all the following quantities

$$\theta_i = \|e_i^\top AA^\dagger\|^2 = e_i^\top AA^\dagger e_i = e_i UU^\top e_i = \|e_i^\top U\|^2.$$

It is known that the leverage scores of a $n \times d$ matrix $A$ with $\mathrm{rank}(A) = r \leq d$ sum to $r$: $\sum_{i=1}^{n} \theta_i = r$. Leverage scores are important in outlier detection, graph sparsification, and numerical linear algebra. We consider the general case where $U$ is not explicitly available, and we only have access to $A$.

To simplify the analysis, we assume that the matrix $A$ has full column rank. The true rank $r$ of $A$ (or the numerical rank, if $A$ is approximately low-rank) as well as a corresponding set of $r$ linearly independent columns of $A$ can be computed in $O(\mathtt{nnz}(A) + d^4/(\epsilon^2\delta))$, with provable approximation guarantees for the leverage scores of the selected column subset. See sections 4 and 5 of [41] for details and [8, 12, 13] for related algorithms and lower bounds.

To use Algorithm 1 to estimate leverage scores, we first need a linear operator that computes $AA^\dagger v$, for an arbitrary vector $v$. Since evaluating $(A^\top A)$ in order to compute its pseudoinverse and ultimately the orthogonal projector $AA^\dagger = A(A^\top A)^\dagger A^\top$ is expensive, we opt for a fast approximate operator. For this we can use standard techniques from the literature. One of the first approximation algorithms for tall-and-skinny leverage scores was proposed in [21]. In [41] it was shown that this algorithm is only efficient for dense matrices, or more specifically for matrices with at least $\omega(\log n)$ nonzeros per row. Given $A \in \mathbb{R}^{n \times d}$ with $n \gg d$ and $\omega(\log n)$ nonzeros per row, the idea consists of approximating the leverage scores of the rows of $A$ with the squared Euclidean row norms of the matrix

$$A(\Pi_1 A)^\dagger \Pi_2.$$

---

[2]Some intermediate steps can be slightly improved using fast matrix multiplication.

Here, $\Pi_1$ is a subspace embedding for $\mathrm{range}(A)$ and $\Pi_2$ is an $\epsilon$-JLT. It can be proved that $(\Pi_1 A)^\dagger$ is in fact an approximate "orthogonalizer" for $A$, a property that we can leverage in our algorithm. Specifically, we apply Algorithm 1 to approximate the Euclidean row norms of $A(\Pi_1 A)^\dagger$, instead of multiplying with $\Pi_2$. This procedure is described in Algorithm 3.

---

**Algorithm 3** Adaptive Leverage Scores Estimation

---

**Input:** $A \in \mathbb{R}^{n \times d}$, with $n \gg d$ and $\omega(\log n)$ nonzeros per row, positive integer $m < d$.
**Output:** Approximate leverage scores $\tilde{\theta}_i \approx \|e_i^\top A A^\dagger\|^2$, $i \in [n]$.
    # Step 1: Construct approximate pseudoinverse operator
1: Construct $\Pi_1$, an $(\epsilon_1, \delta)$-OSE for $\mathrm{range}(A)$.
2: Compute $R$ from a QR factorization of $\Pi_1 A$, i.e. $\Pi_1 A = QR$ and use $R^{-1}$ as a substitute for $(\Pi_1 A)^\dagger$.
    # Step 2: Low-rank approximation
3: Construct two random matrices $S, G \in \mathbb{R}^{d \times m}$ with i.i.d. elements from $\mathcal{N}(0,1)$.         $\triangleright O(dm)$
4: Compute the product $\tilde{S} = R^{-T}(A^\top(A(R^{-1}S)))$.         $\triangleright O(T_{\mathrm{MM}}(A, m))$
5: Compute an orthonormal basis $Q \in \mathbb{R}^{d \times m}$ for $\mathrm{range}(\tilde{S})$ (e.g., via QR).         $\triangleright O(dm^2)$
    # Step 3: Project and compute row norms
6: Compute $\tilde{A} = A(R^{-1}Q)$ and $C = A(R^{-1}G)$.         $\triangleright O(T_{\mathrm{MM}}(A, m))$
7: Compute $\tilde{\Delta} = A(I - QQ^\top)G = C - \tilde{A}(Q^\top G)$.         $\triangleright O(dm^2)$
8: **return** $\mathrm{Alg3}(A, i) = \tilde{\theta}_i = \|(e_i^\top B)\tilde{A}\|^2 + \|(e_i^\top B)\tilde{\Delta}\|^2$, for all $i \in [n]$.         $\triangleright O(nm)$

---

The following theorem gives approximation bounds for the leverage scores returned by Algorithm 3.

**Theorem 3.** *Let $A \in \mathbb{R}^{n \times d}$, $\theta_i = \|e_i^\top A A^\dagger\|^2$ and $\tilde{\theta}_i$ the values returned by Algorithm 3. The following hold:*

$$|\tilde{\theta}_i - \theta_i| \leq (\epsilon_1 + \sqrt{\epsilon_2})\theta_i, \quad and \quad \left| \sum_{i=1}^n \tilde{\theta}_i - d \right| \leq (\epsilon_1 + \epsilon_2)d.$$

*Proof.* Let $\hat{\theta}_i = \|e_i^\top A(\Pi_1 A)^\dagger\|^2$, so that

$$|\tilde{\theta}_i - \theta_i| = |\tilde{\theta}_i - \hat{\theta}_i + \hat{\theta}_i - \theta_i| \leq |\tilde{\theta}_i - \hat{\theta}_i| + |\hat{\theta}_i - \theta_i|.$$

From [21, Lemma 9] it follows that $|\hat{\theta}_i - \theta_i| \leq \frac{\epsilon_1}{1-\epsilon_1}\theta_i$. Subsequently, $\hat{\theta}_i \leq (1 + \frac{\epsilon_1}{1-\epsilon_1})\theta_i = \frac{1}{1-\epsilon_1}\theta_i$. From Theorem 1 we recall that for appropriate $m, l, k$,

$$|\hat{\theta}_i - \tilde{\theta}_i| \leq \sqrt{\epsilon_2}\hat{\theta}_i \leq (\frac{\sqrt{\epsilon_2}}{1-\epsilon_1})\theta_i.$$

Combining all these observations we find that

$$|\tilde{\theta}_i - \theta_i| \leq |\tilde{\theta}_i - \hat{\theta}_i| + |\hat{\theta}_i - \theta_i| \leq \frac{\epsilon_1 + \sqrt{\epsilon_2}}{1 - \epsilon_1}\theta_i \leq 2(\epsilon_1 + \sqrt{\epsilon_2})\theta_i.$$

Rescaling $\epsilon_1$ and $\epsilon_2$ gives the element-wise bounds. For the Frobenius norm bounds we can use similar arguments in combination with Theorem 2. $\qquad\square$

**Complexity and choice of $\epsilon_1, \epsilon_2$.** The complexity of Algorithm 3 can be split into two parts: $(i)$ the complexity of obtaining an $\epsilon_1$-approximate orthonormal basis for $A$, and $(ii)$ the complexity of estimating the row norms of this basis. The complexity of the former has been heavily studied in the literature and depends on the choice of the subspace embedding. For very tall-and-skinny matrices, an efficient construction is to use a combination of a CountSketch [15, 35], a Subsampled Randomized Hadamard Transform (SRHT) [43] and a Gaussian subspace embedding; see e.g. [15]. This provides a sketch $\Pi_1 A$ with dimension $O(d/\epsilon^2) \times d$ in $T(\Pi_1 A)$ time. Computing the QR factorization of the sketch requires $O(d^3/\epsilon^2)$ to obtain $R$. Since $R$ is upper triangular, the computation of $R^{-1}G$ and $R^{-1}Q$ both take $O(d^2 m)$. The products $A(R^{-1}Q)$ and $A(R^{-1}G)$ cost $O(T_{\mathrm{MM}}(A, m))$ each. The last step takes $O(nm)$. The total complexity is

$$O(T(\Pi_1 A) + d^3/\epsilon_1^2 + T_{\mathrm{MM}}(A, m) + nm).$$

To achieve $\epsilon$-accuracy in the Frobenius norm, it suffices to use $m = O(\sqrt{\log(n/\delta)}/\epsilon_2)$. With $\epsilon_1 = \epsilon_2 = \epsilon$, the total complexity becomes

$$O(T(\Pi_1 A) + d^3/\epsilon^2 + T_{\text{MM}}(A, \sqrt{\log(n/\delta)}/\epsilon) + n\sqrt{\log(n/\delta)}/\epsilon).$$

If, instead, we use standard JL projections to estimate the row norms of $AR^{-1}$, the total complexity is

$$O(T(\Pi_1 A) + d^3/\epsilon^2 + T_{\text{MM}}(A, \log(n/\delta)/\epsilon^2) + n\log(n/\delta)/\epsilon^2),$$

to achieve the same Frobenius norm-wise accuracy. For any matrix which is "tall-enough" such that the $O(T_{\text{MM}}(A, \log(n/\delta)/\epsilon^2))$ factor dominates the complexity, Algorithm 3 achieves a quadratic improvement over standard estimators.

## 5 Numerical experiments

Algorithm 1 was implemented in Python using NumPy. We conducted experiments to verify the approximation guarantees and the convergence improvements against standard Gaussian random projections. Following [34], we generated synthetic matrices with decay in the spectrum. Specifically, $d \times d$ matrices $A$, with $d = 5000$, were created as follows. We drew a random orthogonal $d \times d$ matrix $Q$. We then fixed a diagonal $d \times d$ matrix $\Lambda$ which defines the eigenvalues of the matrix. Each element $\Lambda_{i,i}$, $i \in [d]$ is set to $i^{-c}$ for a given $c \geq 0$. The larger the $c$, the faster the spectral decay. We finally constructed the symmetric $A = Q\Lambda Q^\top$ which were used in the numerical experiments. Following [34], we applied four different decay factors, specifically $c = \{0.5, 1, 1.5, 2\}$.

The approximation errors of standard JL projections versus Algorithm 1 are compared in Figure 1. We plot the approximation errors of both methods as the number of samples increases. We plot two types of errors, the maximum element-wise and the Frobenius norm-wise errors

$$\max_{i \in [d]} \frac{|\tilde{x}_i - \|e_i^\top A\|^2|}{\|e_i^\top A\|^2} \quad \text{and} \quad \frac{\left| \tilde{X} - \|A\|_F^2 \right|}{\|A\|_F^2},$$

where $\tilde{x}_i$ are the approximated row norms and $\tilde{X}$ is their sum returned by either Algorithm 1 or standard JL projections. The exact same number of matrix vector queries is used in both methods. Standard JL projections involve only one random matrix $G$, which is multiplies $A$ from the right. $G$ has size $d \times m$, $m$ being the number of samples. In Algorithm 1, on the other hand, $A$ is multiplied four times with a matrix from the right. Therefore, we set $G$, $S$, and $Q$ in Algorithm 1 to have size $d \times m/4$, so that both algorithms are tested with the same number of matrix-vector products. In each plot we illustrate the mean error over 10 independent runs and the standard deviation. Standard JL approximations perform marginally better than Algorithm 1 only for the element-wise errors and only for the matrix with very slow decay. In all other cases, Algorithm 1 performs significantly better.

## 6 Conclusion

We proposed an adaptive algorithm to estimate the Euclidean row norms of a matrix $A$. This algorithm improves standard Johnson-Lindenstrauss estimators in the following aspects: $(i)$ Quadratically less matrix-vector queries are required to achieve the same Frobenius norm-wise accuracy for all matrices; $(ii)$ Asymptotically less matrix-vector queries are needed to achieve the same element-wise accuracy for matrices with decaying spectrum; $(iii)$ At least as accurate element-wise approximations as standard JL are achieved for worst-case input matrices, that is, for matrices with flat spectrum. We also showed how these results can be applied to other important problems, specifically to estimate Euclidean distances between data points, which is related to the fundamental concept of approximate isometries that has many applications in data science, as well as for statistical leverage scores estimations, which are ubiquitous quantities not only in data science and statistics, but also in numerical linear algebra and spectral graph theory.

As future work, several directions can be envisioned. Most prominently, it would be interesting to determine whether the studied techniques can be used to improve Oblivious Subspace Embeddings [40, 44]. Such improvements would have an immediate impact in many problems in NLA, e.g. least squares regression, low-rank approximations and column subset selection. Two other relevant topics concern $(a)$ the possibility to derive lower bounds similar to [34] for Euclidean row norms estimation and $(b)$ to make the algorithms non-adaptive, like the non-adaptive versions of Hutch++ [28, 34] which are based on results from [14], or the Nyström++ of [37].

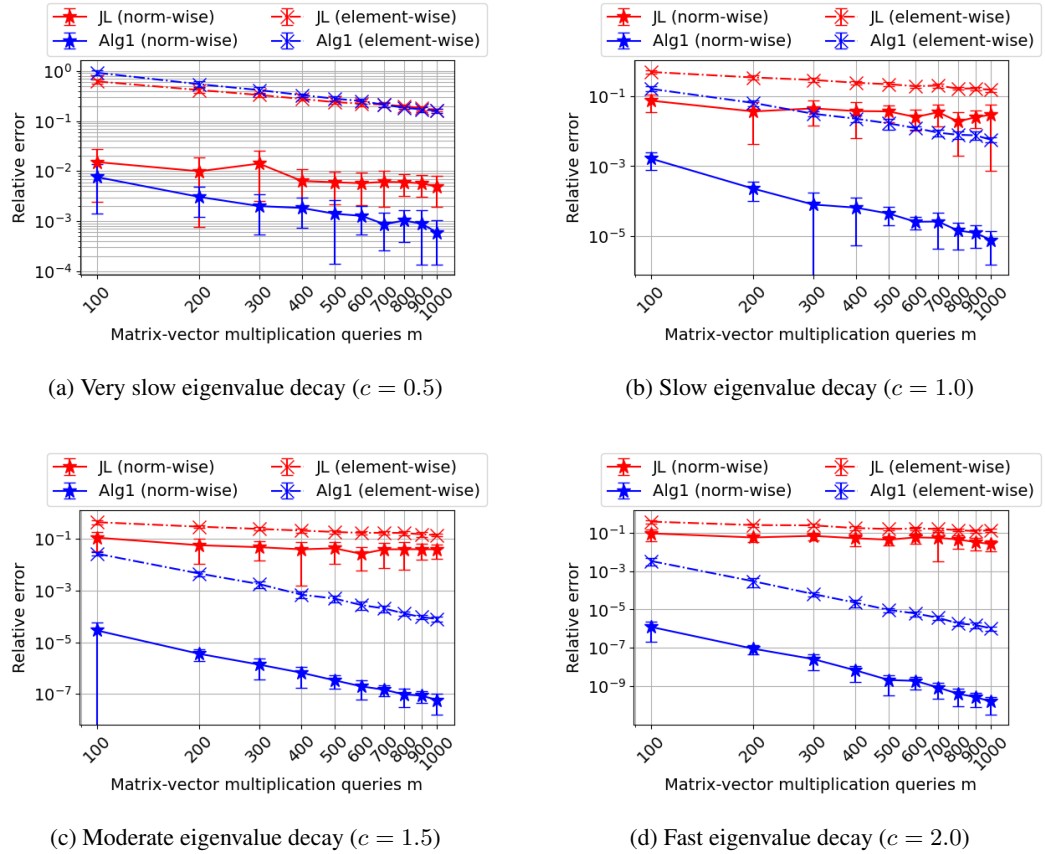

(a) Very slow eigenvalue decay ($c = 0.5$)

(b) Slow eigenvalue decay ($c = 1.0$)

(c) Moderate eigenvalue decay ($c = 1.5$)

(d) Fast eigenvalue decay ($c = 2.0$)

Figure 1: Comparison between the element-wise (dashed curves with "×" marker) and norm-wise (solid curves with "star" marker) relative errors of Algorithm 1 (blue) and standard Gaussian random projections (red) versus number of matrix-vector multiplication queries (x-axis) ran on random matrices with power law spectra. The mean relative error of the approximation averaged over 10 independent runs is plotted. The upper and lower bounds around each curve represent the standard deviation. As expected, for matrices with a very slow decay standard JL projections perform marginally better with respect to the element-wise errors, but Algorithm 1 performs significantly better for all other cases.

## Acknowledgements

The authors would like to thank Cameron Musco for helpful comments.

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
