# I Background in Linear Algebra

In this section we state some elementary results that we will use for our main proofs.

## I.1 Johnson-Lindenstrauss and subspace embeddings

A useful definition for our proofs is the JL moment property, which bounds the moments of the length of $Sx$.

**Definition 3** (JL moment property, [30]). *A distribution $\mathcal{D}$ on matrices $S \in \mathbb{R}^{k \times d}$ has the $(\epsilon, \delta, l)$-JL moment property if given $S \sim \mathcal{D}$, for all $x \in \mathbb{R}^d$ with $\|x\|_2 = 1$ it holds that $\mathbb{E}\left[\left|\|Sx\|^2 - 1\right|^l\right] \leq \epsilon^l \delta$.*

We mention a corollary from [40] which states that JLTs also preserve pairwise angles, which is an important by-product that we will use in our proofs.

**Corollary 2** ([40]). *If $S$ is an $(\epsilon, \delta, n)$-JLT, then for any fixed $n$-element set $V \subset \mathbb{R}^d$, with probability at least $1 - \delta$ it holds that $\langle u, v \rangle - \epsilon \|u\| \|v\| \leq \langle Su, Sv \rangle \leq \langle u, v \rangle + \epsilon \|u\| \|v\|$, for all pairs $u, v \in V$.*

The next Lemma is part of the proof of [44, Lemma 4.2], which we state here as a separate result to save some space from the longer proofs that follow later.

**Lemma 4.** *Let $S$ be a $(\epsilon, \delta)$-OSE for a $d \times k$ matrix $U_k$ with orthonormal columns, $k < d$. Then there exists a $k \times k$ invertible matrix $C$ such that $(SU_k)^\dagger = C^{-1}(SU_k)^\top$ and $\|C^{-1}\| \leq \frac{1}{1-\epsilon}$ with probability at least $1 - \delta$.*

*Proof.* This is part of the proof of [44, Lemma 4.2]. $\qquad\square$

We also repeat here Lemma 4.2 from [44] regarding low-rank approximations as it will be used in upcoming proofs.

**Lemma 5** (Restatement Lemma 4.2 in [44]). *Let $S$ be an $(1/3, \delta)$-OSE for a fixed $k$-dimensional subspace and let $S$ also satisfy the $(\sqrt{\epsilon/k}, \delta, l)$-JL moment property for some $l \geq 2$. Then the rowspace of $SA$, for some $A$, contains a $(1 + \epsilon)$ rank-$k$ approximation to $A$, that is, there exists a $k$-dimensional subspace $M$ within $\mathrm{rowspace}(SA)$ with an orthogonal projector matrix $\Pi_k$ such that*

$$\|A(I - \Pi_k)\|_F^2 \leq (1 + O(\epsilon))\|A - A_k\|_F^2.$$

## I.2 Properties of Gaussian matrices

In this section we specialize the definitions to the case of Gaussian matrices. In our analysis we will use the following restatement of JLTs, addressing the error as a function of the rows of $G$.

**Corollary 3.** *Let $G \in \mathbb{R}^{r \times d}$ with i.i.d. elements from $\mathcal{N}(0, 1/\sqrt{r})$ and $\epsilon \in (0, 1/2)$. For a set $X \subset \mathbb{R}^d$ of $n$ vectors and for $\delta \in (0, 1/2)$, as long as $r > 32 \log(2n/\delta)$, then with probability at least $1 - \delta$ for all $x \in X$ it holds that*

$$\left|\|x\|^2 - \|Gx\|^2\right| \leq \sqrt{\frac{8 \log(2n/\delta)}{r}} \|x\|^2.$$

*Proof.* From Lemma 1 we have that

$$\Pr\left[\left|\|x\|^2 - \|Gx\|^2\right| > \epsilon \|x\|^2\right] \leq 2 \exp\left(-\frac{r(\epsilon^2 - \epsilon^3)}{4}\right).$$

From $\epsilon \in (0, 1/2)$ it follows that

$$\exp\left(-\frac{r(\epsilon^2 - \epsilon^3)}{4}\right) = \exp\left(-\frac{r\epsilon^2(1-\epsilon)}{4}\right) \leq \exp\left(-\frac{r\epsilon^2}{8}\right)$$

As long as $r > 32 \log(2n/\delta)$, we can set $\epsilon = \sqrt{\frac{8 \log(2n/\delta)}{r}}$ and then replace $\epsilon$ in the exponent to obtain

$$\exp\left(-\frac{r(\epsilon^2)}{8}\right) = \exp\left(-\log(2n/\delta)\right) = \delta/(2n).$$

Applying a union bound we find that, as long as $r > 32 \log(2n/\delta)$, for all $x \in X$ simultaneously, it holds that

$$\left| \|x\|^2 - \|Gx\|^2 \right| \leq \sqrt{\frac{8 \log(2n/\delta)}{r}} \|x\|^2$$

with probability at least $1 - \delta$. $\qquad \square$

Next, we also state the required dimension for a scaled Gaussian matrix to satisfy the $(\epsilon, \delta, 2)$-JL moment property and the $(\epsilon, \delta)$-OSE property.

**Lemma 6.** *Let $G$ be a $r \times d$ matrix with i.i.d elements from $\mathcal{N}(0, 1/\sqrt{r})$, and $\epsilon, \delta \in (0, 1/2)$. If $r \geq 2/(\epsilon^2 \delta)$, then $G$ satisfies the $(\epsilon, \delta, 2)$-JL moment property.*

*Proof.* From Definition 3, it must be shown that $\mathbb{E}(\|Gx\|_2^2 - 1)^2 = \mathbb{E}\|Gx\|_2^4 - 2\mathbb{E}\|Gx\|_2^2 + 1 \leq \epsilon^2 \delta$. From the rotation invariance property of the Gaussian distribution, $Gx$ can be replaced with a vector $g \in \mathbb{R}^r$ with i.i.d elements from $\mathcal{N}(0, 1/\sqrt{r})$. We then calculate

$$\mathbb{E}\|g\|_2^2 = \mathbb{E} \sum_{i=1}^{r} |g_i|^2 = \sum_{i=1}^{r} \mathbb{E}|g_i|^2 = \sum_{i=1}^{r} \frac{1}{r} = 1,$$

as well as

$$\begin{aligned} \mathbb{E}\|g\|_2^4 = \mathbb{E}(\sum_{i=1}^{r} |g_i|^2)^2 &= \mathbb{E}(\sum_{i=1}^{r} |g_i|^4) + \mathbb{E} \sum_{i=1}^{r} \sum_{j \neq i} |g_i|^2 |g_j|^2 \\ &= \sum_{i=1}^{r} \mathbb{E}(|g_i|^4) + \sum_{i=1}^{r} \sum_{j \neq i} \mathbb{E}|g_i|^2 |g_j|^2 \\ &= \sum_{i=1}^{r} 3/r^2 + \sum_{i=1}^{r} \sum_{j \neq i} 1/r^2 \\ &= 3/r + (r-1)/r = (r+2)/r. \end{aligned}$$

Putting everything together we obtain

$$(r+2)/r - 2 + 1 \leq \epsilon^2 \delta \Leftrightarrow r \geq 2/(\epsilon^2 \delta).$$

$\qquad \square$

The next theorem from [44] states the required dimension for a Gaussian matrix to provide an $(\epsilon, \delta)$-OSE for a fixed $k$-dimensional subspace.

**Theorem 4** (Thm. 2.3 in [44]). *Let $G \in \mathbb{R}^{r \times d}$ with i.i.d elements from $\mathcal{N}(0, \frac{1}{\sqrt{r}})$, and $\epsilon, \delta \in (0, 1/2)$. If $r = \Theta\left(\frac{k + \log(1/\delta)}{\epsilon^2}\right)$, then $G$ is an $(\epsilon, \delta)$-OSE for any fixed $k$-dimensional subspace.*

### I.3 Basic inequalities

The next lemma gives a bound on powers of logarithms which we will use to simplify the terms in several proofs.

**Lemma 7.** *Let $n \geq 1$ be an integer, and $\delta \in (0, 1/2)$. Then for any constant $c \geq 2$ it holds that*

$$\log(c(n/\delta)) \leq (1 + \log_2 c) \log(n/\delta).$$

*Proof.* $\log(c\frac{n}{\delta}) = \log(2^{\log_2 c}(\frac{n}{\delta})) \leq \log((\frac{1}{\delta})^{\log_2 c}(\frac{n}{\delta})) = \log(\frac{n}{\delta^{1 + \log_2 c}}) \leq \log((\frac{n}{\delta})^{1 + \log_2 c})$. $\qquad \square$

## II    Limitations of low-rank projections

We give a "problematic" example to demonstrate the limitations of low-rank projection based methods. Let $A = I_d$ and assume we want to use a rank-$k$ approximation to estimate the norms of the rows of $A$ for some $k < d$. That is, we want to find a matrix $Q_k$ with $k$ orthonormal columns such that the quantities $\|e_i^\top (A - AQ_kQ_k^\top)\|_2^2$ are small. More formally, $Q_k$ should be a minimizer for the maximum norm of all $i \in [d]$,

$$Q_k = \arg \min_{\substack{Q \in \mathbb{R}^{d \times k} \\ Q^\top Q = I_k}} \max_{i \in [d]} \|e_i^\top (A - AQQ^\top)\|_2^2.$$

Since the maximum is larger than or equal to the average, then for every $Q$ we have that

$$\max_{i \in [d]} \|e_i^\top (A - AQQ^\top)\|^2 \geq \frac{1}{d} \sum_{i=1}^{d} \|e_i^\top (A - AQQ^\top)\|^2 = \frac{\|A - AQQ^\top\|_F^2}{d} = (d-k)/d,$$

where the last equality comes from the fact that $A = I_d$ by assumption and therefore $\|I - QQ^\top\|_F = \sqrt{d-k}$ since $I - QQ^\top$ is an orthogonal projector on a $(d-k)$-dimensional subspace. Subsequently,

$$\min_{\substack{Q \in \mathbb{R}^{d \times k} \\ Q^\top Q = I_k}} \max_{i \in [d]} \|e_i^\top (A - AQQ^\top)\|_2^2 \geq (d-k)/d.$$

If there exists a $Q$ that satisfies this lower bound, then it is also a minimizer. For simplicity let us assume that $k$ divides $d$ exactly. We can construct $Q_k$ as $d/k$ copies of a $k \times k$ orthogonal matrix $H$, that is, $Q_k^\top = \begin{pmatrix} H^\top & H^\top & H^\top & ... & H^\top \end{pmatrix}$. This way $Q_kQ_k^\top$ is a matrix with $d/k$ diagonals in equally spaced positions. We can also scale $Q_kQ_k^\top$ by $k/d$ to ensure that it is an orthogonal projector. All rows of $AQ_kQ_k^\top$ have the same length, equal to $\|e_i^\top Q_kQ_k^\top\|^2 = k/d$. Therefore, $\|e_i^\top (I - Q_kQ_k^\top)\|^2 = 1 - k/d = (d-k)/d$, for all $i \in [d]$, meaning that $Q_k$ is indeed a minimizer.

It is evident that we need $k$ to be almost equal to $d$ in order for this quantity to be small. Specifically, if we want to achieve $\|e_i^\top (I - Q_kQ_k^\top)\|^2 \leq \epsilon$, then we need $(d-k)/d \leq \epsilon \Leftrightarrow d - k \leq \epsilon d \Leftrightarrow k \geq d(1-\epsilon)$. Hence, to obtain a small $\epsilon$-accuracy $k$ must be set almost as large as $d$. We can finally conclude that there exist corner cases where $O(d)$ samples are needed to achieve $\epsilon$-accuracy for element-wise Euclidean norm estimation based on low rank projections.

## III    Projecting rows on randomly chosen subspaces

We state the following main lemma regarding projections on randomly chosen low-rank subspaces.

**Lemma 8** (Oblique projection from $\mathrm{rowspace}(A)$ on $\mathrm{rowspace}(SA^\top A)$)**.** *Let $A \in \mathbb{R}^{n \times d}$, $\epsilon, \delta, \epsilon', \delta' \in (0, 1/2)$ be input parameters, and $S$ have the following properties:*

- *$S \sim \mathcal{D}$, where $\mathcal{D}$ is an $(\epsilon, \delta)$-OSE for any fixed $k$-dimensional subspace;*

- *$S$ is an $(\epsilon', \delta', 2n)$-JLT.*

*Then there exists a rank-$k$ projection matrix $\Pi_k$ within $\mathrm{rowspace}(SA^\top A)$ such that for all $i \in [n]$ simultaneously, it holds that*

$$\|e_i^\top A(I - \Pi_k)\|^2 \leq \|e_i^\top (A - A_k)\|^2 + \frac{\epsilon'}{1 - \epsilon} \frac{\sigma_{k+1}^2(A)}{\sigma_k^2(A)} \|e_i^\top A_k\| \|e_i^\top (A - A_k)\|,$$

*with probability at least $1 - \delta - \delta'$.*

*Proof.* Let $A_k = U_k \Sigma_k V_k^\top$ be the compact SVD of $A_k$. The existence of $\Pi_k$ is proved by construction, that is, we consider the matrix $\Pi_k = V_k(SV_k\Sigma_k^2)^\dagger SA^\top A$. Clearly, this $\Pi_k$ is a rank-$k$ matrix

within $\mathrm{rowspace}(SA^\top A)$. It is also straightforward to verify that it is a projector matrix, since

$$\Pi_k^2 = V_k(SV_k\Sigma_k^2)^\dagger S\overbrace{A^\top A V_k(SV_k\Sigma_k^2)^\dagger SA^\top A}^{=V_k\Sigma_k^2}$$

$$= V_k\overbrace{(SV_k\Sigma_k^2)^\dagger SV_k\Sigma_k^2}^{=I_{k\times k}}(SV_k\Sigma_k^2)^\dagger SA^\top A$$

$$= V_k(SV_k\Sigma_k^2)^\dagger SA^\top A$$

$$= \Pi_k.$$

Clearly, $I - \Pi_k$ is also a projector since $(I - \Pi_k)^2 = I - 2\Pi_k + \Pi_k^2 = I - 2\Pi_k + \Pi_k = I - \Pi_k$. We can then prove the lemma by starting from the left-hand-side of the target inequality. We write

$$\|e_i^\top A(I - \Pi_k)\|^2 = \mathrm{tr}(e_i^\top A(I - \Pi_k)(I - \Pi_k)A^\top e_i)$$

$$= \mathrm{tr}(e_i^\top A(I - \Pi_k)A^\top e_i)$$

$$= \mathrm{tr}(e_i^\top AA^\top e_i) - \mathrm{tr}(e_i^\top A\Pi_k A^\top e_i) \quad \text{(by linearity of the trace)}$$

$$= \mathrm{tr}(A^\top e_i e_i^\top A) - \mathrm{tr}(A^\top e_i e_i^\top A\Pi_k) \quad \text{(by the trace cyclic property)}$$

$$= \mathrm{tr}(A^\top e_i e_i^\top A) - \mathrm{tr}\left(A^\top e_i e_i^\top AV_k(SV_k\Sigma_k^2)^\dagger SA^\top A\right)$$

$$= \mathrm{tr}(A^\top e_i e_i^\top A) - \mathrm{tr}\left(A^\top e_i e_i^\top AV_k(SV_k\Sigma_k^2)^\dagger S\overbrace{(V_k\Sigma_k^2 V_k^\top + \bar{V}_k\bar{\Sigma}_k^2\bar{V}_k^\top)}^{=A^\top A}\right)$$

$$= \mathrm{tr}(A^\top e_i e_i^\top A) - \mathrm{tr}\left(A^\top e_i e_i^\top AV_k\overbrace{(SV_k\Sigma_k^2)^\dagger SV_k\Sigma_k^2}^{=I_{k\times k}}V_k^\top\right) - \ldots$$

$$\ldots - \mathrm{tr}\left(A^\top e_i e_i^\top AV_k(SV_k\Sigma_k^2)^\dagger S\bar{V}_k\bar{\Sigma}_k^2\bar{V}_k^\top\right)$$

$$= \mathrm{tr}(A^\top e_i e_i^\top A) - \mathrm{tr}(A^\top e_i e_i^\top AV_k V_k^\top) - \ldots$$

$$\ldots - \mathrm{tr}\left(A^\top e_i e_i^\top AV_k(SV_k\Sigma_k^2)^\dagger S\bar{V}_k\bar{\Sigma}_k^2\bar{V}_k^\top\right)$$

$$= \mathrm{tr}\left(A^\top e_i e_i^\top A(I - V_k V_k^\top)\right) - \mathrm{tr}\left(A^\top e_i e_i^\top AV_k(SV_k\Sigma_k^2)^\dagger S\bar{V}_k\bar{\Sigma}_k^2\bar{V}_k^\top\right)$$

$$= \|e_i^\top(A - A_k)\|^2 - \mathrm{tr}\left(A^\top e_i e_i^\top AV_k(SV_k\Sigma_k^2)^\dagger S\bar{V}_k\bar{\Sigma}_k^2\bar{V}_k^\top\right)$$

$$= \|e_i^\top(A - A_k)\|^2 - e_i^\top AV_k(SV_k\Sigma_k^2)^\dagger S\bar{V}_k\bar{\Sigma}_k^2\bar{V}_k^\top A^\top e_i \quad \text{(by cyclic property)}$$

$$\leq \|e_i^\top(A - A_k)\|^2 + \left|e_i^\top AV_k(SV_k\Sigma_k^2)^\dagger S\bar{V}_k\bar{\Sigma}_k^2\bar{V}_k^\top A^\top e_i\right|$$

$$\leq \|e_i^\top(A - A_k)\|^2 + \left|e_i^\top AV_k\Sigma_k^{-2}(SV_k)^\dagger S\bar{V}_k\bar{\Sigma}_k^2\bar{V}_k^\top A^\top e_i\right|$$

$$= \|e_i^\top(A - A_k)\|^2 + \left|e_i^\top AV_k\Sigma_k^{-2}C^{-1}V_k^\top S^\top S\bar{V}_k\bar{\Sigma}_k^2\bar{V}_k^\top A^\top e_i\right|,$$

where the last equality comes from Lemma 4. To conclude the proof, it suffices to determine a bound for the rightmost term. To do this, notice that $|e_i^\top AV_k\Sigma_k^2 C^{-1}V_k^\top S^\top S\bar{V}_k\bar{\Sigma}_k^2\bar{V}_k^\top A^\top e_i|$ is the absolute value of the inner product $\langle S(V_kC^{-1}V_k^\top A^\top)e_i, S(\bar{V}_k\bar{V}_k^\top A^\top)e_i\rangle$, which can be written in a simplified form as $\langle Sx_k, S\bar{x}_k\rangle$. Therefore, we must bound the inner products between vectors from a specific set. We consider the following set of vectors:

$$V = \left\{e_i^\top AV_k\Sigma_k^2 C^{-1}V_k^\top | i \in [n]\right\} \bigcup \left\{e_i^\top A\bar{V}_k\bar{\Sigma}_k^2\bar{V}_k^\top | i \in [n]\right\}.$$

Clearly, $|V| = 2n$. Recall that by assumption $S$ is an $(\epsilon', \delta', 2n)$-JLT. Therefore, by Corollary 2, $S$ preserves all inner products between all pairs of vectors in $V$ with probability at least $1 - \delta'$. We have

$$\left| e_i^\top A V_k \Sigma_k^{-2} C^{-1} V_k^\top S^\top S \bar{V}_k \bar{\Sigma}_k^2 \bar{V}_k^\top A^\top e_i \right| \le e_i^\top A V_k \Sigma_k^{-2} C^{-1} \overbrace{V_k^\top \bar{V}_k}^{\text{orthogonal}} \bar{\Sigma}_k^2 \bar{V}_k^\top A^\top e_i + \ldots$$

$$\ldots + \epsilon' \| e_i^\top A V_k \Sigma_k^{-2} C^{-1} V_k^\top \| \| \bar{V}_k \bar{\Sigma}_k^2 \bar{V}_k^\top A^\top e_i \|$$

$$= \epsilon' \| e_i^\top A V_k \Sigma_k^2 C^{-1} V_k^\top \| \| \bar{V}_k \bar{\Sigma}_k^2 \bar{V}_k^\top A^\top e_i \|$$

$$\le \epsilon' \| e_i^\top A V_k \| \| \Sigma_k^{-2} \| \| C^{-1} \| \| \bar{\Sigma}_k^2 \| \| e_i^\top (A - A_k) \|$$

$$\le \frac{\epsilon'}{1 - \epsilon} \frac{\sigma_{k+1}^2(A)}{\sigma_k^2(A)} \| e_i^\top A_k \| \| e_i^\top (A - A_k) \|.$$

Combining the two inequalities we find that

$$\| e_i^\top A (I - \Pi_k) \|^2 \le \| e_i^\top (A - A_k) \|^2 + \frac{\epsilon'}{1 - \epsilon} \frac{\sigma_{k+1}^2(A)}{\sigma_k^2(A)} \| e_i^\top A_k \| \| e_i^\top (A - A_k) \|.$$

$\square$

# A Proofs for Section 2 (Analysis of Algorithm 1)

**Lemma 2.** *Let $A \in \mathbb{R}^{n \times d}$. If we use Algorithm 1 with $m$ matrix-vector queries to estimate the Euclidean lengths of the rows $e_i^\top A$, $i \in [n]$, then as long as $m \geq l \geq 32 \log(4n/\delta)$ it holds that*

$$\left| \tilde{x}_i - \|e_i^\top A\|^2 \right| \leq \sqrt{\frac{8 \log(\frac{2n}{\delta})}{l}} \|e_i^\top A (I - QQ^\top)\|^2, \text{ for all } i \in [n],$$

*with probability at least $1 - \delta$ for all $i \in [n]$ simultaneously.*

*Proof.* We start by noting that

$$\begin{aligned}
|\tilde{x}_i - \| e_i^\top A\|^2| &= \left| \|e_i^\top AQ\|^2 + \|e_i^\top A(I - QQ^\top)G\|^2 - \|e_i^\top A\|^2 \right| \\
&= \left| \|e_i^\top AQ\|^2 + \|e_i^\top A(I - QQ^\top)G\|^2 - \|e_i^\top AQQ^\top\|^2 - \|e_i^\top A(I - QQ^\top)\|^2 \right| \\
&= \left| \|e_i^\top A(I - QQ^\top)G\|^2 - \|e_i^\top A(I - QQ^\top)\|^2 \right|.
\end{aligned}$$

Since $G$ is a scaled Gaussian matrix with $l$ columns, we can use Corollary 3, which implies that as long as $l \geq 32 \log(4n/\delta)$ it holds that

$$\left| \|e_i^\top A(I - QQ^\top)G\|^2 - \|e_i^\top A(I - QQ^\top)\|^2 \right| \leq \sqrt{\frac{8 \log(\frac{2n}{\delta})}{l}} \|e_i^\top A(I - QQ^\top)\|^2, \quad (2)$$

with probability at least $1 - \delta$ for all $i \in [n]$. $\qquad\square$

**Corollary 1** (Projection on $\mathrm{rowspace}(SA^\top A)$). *Let $\delta \in (0, \frac{1}{2})$, $\bar{A}_k = A - A_k$, and $S$ be such that*

*(i) $S \sim \mathcal{D}$, where $\mathcal{D}$ is an $(1/3, \delta)$-OSE for any fixed $k$-dimensional subspace;*

*(ii) $S$ is a $(1/3, \delta, 2n)$-JLT.*

*If $Q$ is a matrix that forms an orthonormal basis for $\mathrm{rowspace}(SA^\top A)$, then, with probability at least $1 - 2\delta$, for all $i \in [n]$ simultaneously, it holds that*

$$\|e_i^\top A(I - QQ^\top)\|^2 \leq \|e_i^\top (\bar{A}_k)\|^2 + \frac{1}{2} \frac{\sigma_{k+1}^2(A)}{\sigma_k^2(A)} \|e_i^\top A_k\| \|e_i^\top \bar{A}_k\| \leq \frac{3}{2} \|e_i^\top A\| \|e_i^\top \bar{A}_k\|.$$

*Proof.* From Lemma 8 we have that inside $\mathrm{rowspace}(SA^\top A)$ there exists a subspace with the desired properties. An orthogonal projection on that subspace via $QQ^\top$ is sufficient to satisfy the results of Lemma 8 for $\epsilon = \epsilon' = 1/3$. The second part comes from the fact that $\|e_i^\top(A - A_k)\| \leq \|e_i^\top A\|$ and $\|e_i^\top A_k\| \leq \|e_i^\top A\|$, as well as the fact that $\sigma_{k+1}(A) \leq \sigma_k(A)$. With a union bound we have that both Lemma 4 and Corollary 2 hold at the same time with probability at least $1 - 2\delta$. $\qquad\square$

**Theorem 1.** *Let $A \in \mathbb{R}^{n \times d}$ and $n \geq d$. If we use Algorithm 1 with $m$ matrix-vector queries to estimate the Euclidean lengths of the rows of $A$, then there exists a global constant $C$ such that, as long as*

*(i) $m \geq l \geq O(\log(n/\delta))$, such that $G$ satisfies Lemma 1 and $S$ forms an $(1/3, \delta, 2n)$-JLT,*

*(ii) $m \geq O(k + \log(1/\delta))$, such that $S$ forms an $(1/3, \delta)$-OSE for a $k$-dimensional subspace,*

*then it holds that*

$$\left| \tilde{x}_i - \|e_i^\top A\|^2 \right| \leq C \sqrt{\frac{\log(\frac{n}{\delta})}{l}} \|e_i^\top (A - A_k)\| \|e_i^\top A\| \leq C \sqrt{\frac{\log(\frac{n}{\delta})}{lk}} \|A_k\|_F \|e_i^\top A\|,$$

*for all $i \in [n]$ with probability at least $1 - 3\delta$.*

*Proof.* For the first inequality, we start directly from Lemma 2, which gives a first bound for the approximated values. We can then use Corollary 1 to bound the rightmost term. By assumption, $S$ satisfies the conditions of Corollary 1, which implies that

$$\|e_i^\top A(I - QQ^\top)\| \leq \frac{3}{2} \|e_i^\top A\| \|e_i^\top (A - A_k)\|$$

with probability at least $1 - 2\delta$. Combining this with Equation (2) we have that

$$\left|\|e_i^\top A(I - QQ^\top)G\|^2 - \|e_i^\top A(I - QQ^\top)\|^2\right| \leq \sqrt{\frac{8\log(\frac{2n}{\delta})}{l}} \cdot \frac{3}{2}\|e_i^\top A\|\|e_i^\top(A - A_k)\|.$$

We then recall Lemma 7 to move the constants outside the logarithms. There are three random events and each of them fails with probability at most $\delta$. Therefore, by taking a union bound we find that the algorithm succeeds with probability at least $1 - 3\delta$. The last inequality of the theorem comes from Lemma 3, which gives bounds for $\|e_i^\top(A - A_k)\|$. $\qquad\square$

**Theorem 2.** *In Algorithm 1, for some absolute constants $c, C$, if $l > c\log(1/\delta)$, it holds that*

$$\left|\tilde{X} - \|A\|_F^2\right| \leq C\sqrt{\frac{\log(\frac{1}{\delta})}{lk}}\|A\|_F^2,$$

*where $\tilde{X}$ is the sum of the returned approximations. For $l = k = O\left(\frac{\sqrt{\log(\frac{1}{\delta})}}{\epsilon}\right)$, where $\epsilon \in (0, 1/2)$, setting $m \geq O(k/\delta + \log(\frac{1}{\delta}))$, it follows that*

$$\left|\tilde{X} - \|A\|_F^2\right| \leq \epsilon\|A\|_F^2.$$

*Proof.* The proof is a direct application of [34, Theorem 3.1]. We state it for completeness. Let $\tilde{A}, \Delta, Q, G, S$ be as in Algorithm 1, that is, $\tilde{A} = AQQ^\top$, $\Delta = A(I - QQ^\top)$, and $Q$ is an orthonormal basis for $\mathrm{range}(A^\top AS)$. Because

$$\tilde{X} = \sum_{i=1}^d \tilde{x}_i = \sum_{i=1}^d \left(\|e_i^\top \tilde{A}\|^2 + \|e_i^\top \Delta G\|^2\right) = \|\tilde{A}\|_F^2 + \|\Delta G\|_F^2,$$

we can then write

$$
\begin{aligned}
\left|\tilde{X} - \|A\|_F^2\right| &= \left|\|\tilde{A}\|_F^2 + \|\Delta G\|_F^2 - \|A\|_F^2\right| \\
&= \left|(\|AQQ^\top\|_F^2 + \|A(I - QQ^\top)G\|_F^2) - (\|AQQ^\top\|_F^2 + \|A(I - QQ^\top)\|_F^2)\right| \\
&= \left|\|A(I - QQ^\top)G\|_F^2 - \|A(I - QQ^\top)\|_F^2\right| \\
&= \left|\mathrm{tr}\left(G^\top(I - QQ^\top)A^\top A(I - QQ^\top)G\right) - \mathrm{tr}\left((I - QQ^\top)A^\top A(I - QQ^\top)\right)\right|.
\end{aligned}
$$

This is identical to using the Hutch++ Algorithm [34] on $A^\top A$ instead of $A$. Note that $A^\top A$ is always symmetric and positive semi-definite. It can be shown that the following two conditions hold, which allows to apply [34, Theorem 3.1]:

$$\mathrm{tr}(A^\top A) = \mathrm{tr}(\tilde{A}^\top \tilde{A}) + \mathrm{tr}(\Delta^\top \Delta), \tag{3}$$

and

$$\|\Delta^\top \Delta\|_F \leq 2\|A^\top A - (A^\top A)_k\|_F. \tag{4}$$

The first condition, Equation 3, is straightforward to prove

$$
\begin{aligned}
\mathrm{tr}(\tilde{A}^\top \tilde{A}) + \mathrm{tr}(\Delta^\top \Delta) &= \mathrm{tr}(QQ^\top A^\top AQQ^\top) + \mathrm{tr}((I - QQ^\top)A^\top A(I - QQ^\top)) \\
&= \mathrm{tr}(A^\top AQQ^\top) + \mathrm{tr}(A^\top A(I - QQ^\top)) \text{ (by the trace cyclic property)} \\
&= \mathrm{tr}(A^\top A(QQ^\top + I - QQ^\top)) \\
&= \mathrm{tr}(A^\top A).
\end{aligned}
$$

For the second condition, we have

$$
\begin{aligned}
\|\Delta^\top \Delta\|_F = \|(I - QQ^\top)A^\top A(I - QQ^\top)\|_F &\leq \|I - QQ^\top\|_2\|A^\top A(I - QQ^\top)\|_F \\
&= \|A^\top A(I - QQ^\top)\|_F.
\end{aligned}
$$

From Lemma 5, we know that as long as $Q$ has $\Omega(k/\delta + \log(1/\delta))$ columns, then

$$\|A^\top A(I - QQ^\top)\|_F \leq 2\|A^\top A - (A^\top A)_k\|_F$$

holds with probability at least $1-\delta$. Therefore, both conditions of [34, Theorem 3.1] are satisfied with probability at least $1 - \delta$, which implies that there exist constants $c, C$ such that, if $l > c\log(1/\delta)$, the quantity

$$Z = \text{tr}(\tilde{A}^\top \tilde{A}) + \text{tr}(G^\top \Delta^\top \Delta G) = \|\tilde{A}\|_F^2 + \|\Delta G\|_F^2 = \tilde{X} \tag{5}$$

satisfies:

$$\left| Z - \text{tr}(A^\top A) \right| \leq 2C\sqrt{\frac{\log(1/\delta)}{kl}} \cdot \text{tr}(A^\top A)$$
$$\Leftrightarrow$$
$$\left| \tilde{X} - \|A\|_F^2 \right| \leq 2C\sqrt{\frac{\log(1/\delta)}{kl}} \cdot \|A\|_F^2.$$

With a trivial union bound both events of the proof hold with probability at least $1 - 2\delta$. Rescaling $\delta$ concludes the proof. $\square$

## B  Proofs for Section 3 (Euclidean distances)

**Theorem 5.** *Let $A \in \mathbb{R}^{t \times d}$, $t \geq d$, for which we want to estimate the distances between pairs of rows. Let $B \in \mathbb{R}^{n \times t}$ be a matrix such that each row of $B$ is equal to $(e_i - e_j)^\top$, meaning that $(e_i - e_j)^\top A$ gives the distance vector between the $i$-th and the $j$-th row of $A$ which we want to estimate. Let $M = BA$. If we apply Algorithm 2 with $m$ matrix-vector queries to estimate the Euclidean lengths of rows of $M$ by the values $\tilde{x}_i, i \in [n]$, then there exists a global constant $C$ such that, as long as*

*(i) $m \geq l \geq O(\log(n/\delta))$, such that $G$ satisfies Lemma 1 and $S$ forms an $(1/3, \delta, 2n)$-JLT,*

*(ii) $m \geq O(k + \log(1/\delta))$, such that $S$ forms an $(1/3, \delta)$-OSE for a $k$-dimensional subspace,*

*then it holds that*

$$\left| \tilde{x}_i - \|e_i^\top M\|^2 \right| \leq C\sqrt{\frac{\log(\frac{n}{\delta})}{l}} \|e_i^\top (M - M_k)\| \|e_i^\top M\| \leq C\sqrt{\frac{\log(\frac{n}{\delta})}{lk}} \|M_k\|_F \|e_i^\top M\|,$$

*for all $i \in [n]$, with probability at least $1 - 3\delta$. In addition, for some absolute constants $c, C$, if $l > c\log(1/\delta)$, it holds that*

$$\left| \tilde{X} - \|M\|_F^2 \right| \leq C\sqrt{\frac{\log(1/\delta)}{lk}} \|M\|_F^2,$$

*where $\tilde{X}$ is the sum of the approximations. For $l = k = O\left( \frac{\sqrt{\log(\frac{1}{\delta})}}{\epsilon} \right)$, where $\epsilon \in (0, 1/2)$, setting $m \geq O(k/\delta + \log(\frac{1}{\delta}))$, it follows that*

$$\left| \tilde{X} - \|M\|_F^2 \right| \leq \epsilon \|M\|_F^2.$$

*Proof.* The bounds are a direct application of Theorems 1 and 2 on $BA$ instead of $A$. $\square$

## References for the Appendix

[34] Raphael A. Meyer, Cameron Musco, Christopher Musco, and David P. Woodruff. Hutch++: Optimal stochastic trace estimation. In *Symposium on Simplicity in Algorithms*, pages 142–155. SIAM, January 2021.