# OpenReview forum: "Approximate Euclidean lengths and distances beyond Johnson-Lindenstrauss"
_NeurIPS.cc/2022/Conference — NeurIPS 2022 Accept_

### Official Review · Reviewer_Bucs · 2022-07-07

**Rating:** 5
**Confidence:** 2
**Soundness:** 4 excellent
**Presentation:** 2 fair
**Contribution:** 3 good

**Summary:**

For the Johnson Lindenstrauss transform we can embed $n$ points in $O(\log(n)\epsilon^{-2})$ dimensional space such that we have at most $\epsilon$ distortion. This paper instead look at the problem of having an embedding that only preserves the norm of the data points instead of the metric. The paper does this so that the dimension of the new space only has an $\epsilon^{-1}$ dependence. The goal is that doing it via such an embedding is faster than just computing the norm of the row.

The paper then presents two applications including computing the metric.

**Questions:**

So for context, the reason we like the JL transform is because we can compute it quickly (i.e. faster than doing regular matrix multiplication) [1]. However, the key idea there is the fact that we can do the transform using a Fourier transform and sparse matrices.

However, in this paper, we are using a Gaussian matrix. So the matrix multiplication for $AS$ in Algorithm 1 step 2 is multiplying a $n \times d$ matrix by a $d \times m$ matrix where $m \le d \le n$. My understanding is that the best rate for this is still $O(m^{\omega-2}dn)$ where $\omega \ge 2$ is the exponent for square matrix multiplication (i.e. multiply two $n\times n$ matrices in time $O(n^\omega)$. However, that is old. Looking for more new results I found [3] which shows that for multiplying an $n \times n^\alpha$ matrix with a $n^\alpha \times n$ matrix can be done in $O(n^{2+\epsilon})$ for any $\epsilon > 0$. Hence I am not sure if that matrix multiplication can be done faster than $O(nd)$ which is the time taken to compute the true norms.

Even otherwise with step taking $O(nm^2)$ we need $m \le \sqrt{d}$.


[1] THE FAST JOHNSON–LINDENSTRAUSS TRANSFORM AND APPROXIMATE NEAREST NEIGHBORS∗
NIR AILON† AND BERNARD CHAZELLE‡
[2] Knight, Philip A., Fast rectangular matrix multiplication and QR decomposition, Linear Algebra Appl. 221, 69-81 (1995). ZBL0827.65044.
[3]  Improved Rectangular Matrix Multiplication using Powers of the Coppersmith-Winograd Tensor by François Le Gall† Florent Urrutia



**Limitations:**

Yes.

**Strengths And Weaknesses:**

**Post Rebuttal**

I thank the authors for their response. Their response does help clarify my questions about applications and the matrix multiplication.

In light of this I raise my score.

**Strengths**

I think the problem is very interesting and I like the proof techniques as well.

The algorithm is theoretically motivated and the results are theoretically proved.

The improvement over the JL is empirically demonstrated.

**weaknesses**

While the paper does present two applications. I am left unclear as to where this problem shows up in practice (even though the problem is interesting).

The cost of matrix multiplication has not been discussed and this is crucial to the viability of the method.

I am not sure if this the best fit for a the machine learning community. Maybe the theoretical CS community might appreciate this more (SODA, FOCS, STOC etc).

---

> ### Author Response · Authors · 2022-08-01
> **Review response 1/3**
>
> We would like to thank the reviewer for their effort and for the detailed comments, and we are very happy that they liked the problem and our techniques. We provide answers to specific questions below.
>
> **Q:**
> So for context, the reason we like the JL transform is because we can compute it quickly (i.e. faster than doing regular matrix multiplication) [1]. However,  ...
>
> ...
>
> [1] THE FAST JOHNSON–LINDENSTRAUSS TRANSFORM AND APPROXIMATE NEAREST NEIGHBORS NIR AILON† AND BERNARD CHAZELLE [2] Knight, Philip A., Fast rectangular matrix multiplication and QR decomposition, Linear Algebra Appl. 221, 69-81 (1995). ZBL0827.65044. [3] Improved Rectangular Matrix Multiplication using Powers of the Coppersmith-Winograd Tensor by François Le Gall Florent Urrutia
>
> **A:**
> These are all valid comments that are raised in this question, we will clarify our case as best as possible below. We break it in two parts.
>
> 1) As a general comment, we highlight again that we consider the matrix-vector query model. I.e., the matrix $A$ is not expected to be explicitly available, but we only have access to an operator that computes the product $Ax$ for some vector $x$. E.g., this is the case when someone has a matrix $A$ and wants to do computations with a function of $A$, e.g. $f(A)$. $f(A)$ is not explicitly available. Depending on how complex the function is, it might require $O(n^\omega)$ operations (or even more) to compute it explicitly. Moreover, if $A$ is sparse, $f(A)$ might be dense and therefore it might not even fit in memory. As a seminal example, consider the case where we have an adjacency matrix $A$ of a graph and we need to compute the effective resistances of the edges. The effective resistances can be found in the diagonal of the Laplacian pseudoinverse. The pseudoinverse requires $O(n^\omega)$ operations to compute. But, in their seminal work, Spielman and Srivastava (see ref. [R1] below) showed that one can use JL random projections in combination with a fast Laplacian solver to approximate the effective resistances of a graph in nearly linear time (!!!). This result cannot be achieved e.g. by using fast rectangular matrix multiplications. Many other matrix functions arise in applications.

---

> > ### Author Response · Authors · 2022-08-01
> > **Review response 2/3**
> >
> >
> > 2) Regarding the cost of computing the row norms directly vs using random projections (either JL or our work). If the matrix is explicitly available, one can directly compute the row norms in $O(nd)$. But as already mentioned above, this is not the case in the matrix-vector-query model, i.e. we do not have access to the matrix. Some more detailed examples are the following:
> >     - Consider the case of estimating leverage scores, which is detailed in Section 4. We have the original input matrix $A$, and we have computed an $R$ such that $AR^{-1}$ has (approximately) orthonormal columns. How does one compute the row norms of $AR^{-1}$? If we compute explicitly the matrix $AR^{-1}$, this costs $O(nd^2)$ if $R$ is $d\times d$. This is much larger than $O(nd)$ which is required for the row norms in the subsequent step. However, if we use a Gaussian JL projection $G$ with $m$ columns we can compute the row norms of $A(R^{-1}G)$. In this case $B=R^{-1}G$ takes $O(d^2m)$ to compute, $m\ll d$. In the second step, $AB$ takes $O(ndm)$ to compute, much less than $O(nd^2)$. Even if $G$ was a sparse or Fourier projection, the total cost would still be dominated by $O(ndm)$, therefore, sparse/structured projections provide no benefit at all. Gaussian projections are more than sufficient. (Note: we did not use fast matrix multiplication in this paragraph but it can be modified accordingly, nothing changes).
> >     - Consider the case where one has a square matrix $n\times n$ (e.g. $n=d$) and we want to approximate a matrix function by a polynomial, e.g. like the famous Kernel Polynomial Method, and consider we want to compute the row norms of that matrix function. Let's say we have the polynomial $5A^2 + 3A$ (arbitrary example). In order to evaluate the $A^2$ term, we need $O(n^\omega)$. If, on the other hand, we use again a Gaussian random projection $G$ with $m$ columns, we can approximate the row norms by the row norms of $(5A^2+3A)G$. This takes $O(n^2m^{\omega-2})$. Again, just like the previous example, we have absolutely no benefit in using a sparse/Fourier matrix instead of a Gaussian matrix, because if we want to compute the matrix $A^2G$, we compute it as $A(AG)$. $AG$ will be cheap, but thereafter $A(AG)$ will still cost $O(n^2m^{\omega-2})$.
> >
> > We hope that this clarifies that Gaussian projections are well justified, and in most cases, it is all one needs. If there are some very specific applications with special structure one can consider swapping the Gaussian matrix with another type of random projection. Nothing changes in our analysis regarding the approximation guarantees, as long as it satisfies the JL-moment-property and the other required properties. We emphasize on this once again: all of our results are stated as structural results. We do not explicitly need Gaussian matrices. We need matrices that satisfy specific properties, e.g. the JL-moment-property, the $\epsilon$-JLT property, the Subspace Embedding property, etc.
> > CountSketch, SRHT, SRFT, OSNAP, they are all excellent candidates to use. Nothing will change in our analysis if they satisfy the required properties.

---

> > > ### Author Response · Authors · 2022-08-01
> > > **Review response 3/3**
> > >
> > > **Q:**
> > > While the paper does present two applications. I am left unclear as to where this problem shows up in practice (even though the problem is interesting). The cost of matrix multiplication has not been discussed and this is crucial to the viability of the method. I am not sure if this the best fit for a the machine learning community. Maybe the theoretical CS community might appreciate this more (SODA, FOCS, STOC etc).
> > >
> > > **A:**
> > > We have highlighted four very successful applications of JL-like random projections in machine learning, specifically, references [2, 6, 16, 11] (see the Introduction also for other applications in other domains). In machine learning, a seminal example where this can be applied is that one can can derive fast approximate solution even for hard problems like $k$-means clustering. In general, we are also not fully aware of a specific application where the ultimate goal is to compute the Euclidean lengths or the distances  per-se, but they are a fundamental building block for a plethora of other applications, especially when dimensionality reduction is crucial. E.g., random projections gave rise to the so-called subspace embeddings (cf. Sarlós [R2]) which revolutionized numerical linear algebra. We do believe that dimensionality reduction is in general highly relevant to the conference and that our work is well within the context.
> > >
> > > Regarding the cost of matrix multiplication, since $A$ is given as an operator in the matrix-vector-query model, we cannot know beforehand the cost of multiplying $A$ with a vector. It depends on the application, therefore, we decided to use the generic term $T_{MM}(A,m)$ to denote the complexity of this operation.
> > >
> > > **Conclusion:**
> > > In conclusion, we hope that we were able to clarify all the concerns raised by the reviewer. If there are further specific questions we might be able to discuss them in the next phase of the review process. We would also be willing to adapt possible modifications that might be suggested in a final revision if the manuscript gets accepted. In any case, we would like to thank them again for their effort and for providing actual constructive comments and criticism.
> > >
> > > **References:**
> > >
> > > [R1] Spielman, Daniel A., and Nikhil Srivastava. "Graph sparsification by effective resistances." SIAM Journal on Computing 40.6 (2011): 1913-1926.
> > >
> > > [R2] Sarlos, Tamas. "Improved approximation algorithms for large matrices via random projections." 2006 47th annual IEEE symposium on foundations of computer science (FOCS'06). IEEE, 2006.

---

### Official Review · Reviewer_FGjt · 2022-07-11

**Rating:** 7
**Confidence:** 3
**Soundness:** 3 good
**Presentation:** 3 good
**Contribution:** 3 good

**Summary:**

The authors give an algorithm to approximate the pairwise Euclidean distances of row vectors of any given matrix. This algorithm improves the bounds of typical Johnson-Lindenstrauss estimators in a few specific settings:
- When achieving the same Frobenius norm-wise accuracy, less matrix-vector queries are required;
- When achieving the same element-wise accuracy less matrix-vector queries are required;
- The algorithm given is as least as accurate as the standard Johnson-Lindenstrauss techniques, when given matrices with flat spectrums.

**Questions:**

Suggestion: improve notation, try to include more details

**Limitations:**

No potentially negative societal impact in sight.

**Strengths And Weaknesses:**

The paper is well written and structured. The algorithms are well presented, but the authors tend to omit details. The main results, that is Theorems 1,2,3 and 5, show that the techniques of the authors are improvements of the Johnson-Lindenstrauss techniques for their specific use-case. The presentation of proofs is overall good, with a few minor inconveniences. The proofs, that are omitted from the paper and are part of the Appendix, are at least sketched, so the reader gets an idea of the proof. Also enough text is given to follow the structure of the paper.

Because of the notation, sometimes the math is hard to read/follow (see the proof sketch of Corollary 1 for example, also Alg1 as placeholder for the output is suboptimal).

The paper is well written, besides some minor issues with the notation. The result is interesting and a real improvement for the cases presented.

---

> ### Author Response · Authors · 2022-08-01
> **Review response**
>
> We would like to thank the reviewer for their effort to truly understand the results and the novelty of this work and for their kind words. Regarding the notation, indeed, initially we thought that using Alg1$(A)$ might have been easier to follow than, for example, $\tilde X$ or so, but in the end this was probably not the case. We are willing to change it accordingly in a revision if the paper gets accepted, including also other specific suggestions that the reviewer might have to improve the presentation.

---

> > ### Author Response · Authors · 2022-08-08
> > **Further Questions**
> >
> > As the deadline for the Reviewer-Author discussion phase approaches, we were wondering if the reviewer has any further questions/comments that we could respond to.

---

> > > ### Comment · Reviewer_FGjt · 2022-08-08
> > > **Response to the Authors**
> > >
> > > Thank you! I don't have any further questions or comments.

---

### Official Review · Reviewer_Lczq · 2022-07-11

**Rating:** 7
**Confidence:** 3
**Soundness:** 3 good
**Presentation:** 2 fair
**Contribution:** 3 good

**Summary:**

The authors provide a dimensionality reduction techniques that is more optimized towards certain data than the traditional JL. Specifically speaking, while JL is known to be optimal when dealing with data with no decay, it is not the optimal method when dealing with datasets that admit exponential or even linear decay. In this paper, the authors provide a technique inspired by the Hutch++ algorithm that deals with such instances of data.

**Questions:**

The results stated in the paper seem sound and are contributing. However, a valid question that usually pops up among practitioners who aim to use dimensionality reduction to boost the performance of their work, are matrices with linear decay/exponential decay common in the field of data science, i.e., how often we get to actually deal with such data that admits such properties?

In addition, is it possible to use some variant of your analysis when dealing with lewis weights?

Finally, please make sure that the main manuscript is at most 9 pages, while the current version of the paper can be refered to as the supplementary material.

**Limitations:**

The authors clearly stated the limitations of their work.

**Strengths And Weaknesses:**

* Strengths:
    1) The paper seems sound theoretically, which hinges upon using delicate and interesting lemmata and theorems.
    2) The experimental section (regardless of using only synthetic data), shows that JL is indeed not optimal for the case where data admits linear/exponential decay.

* Weaknesses:
    1) The writing of the paper is somewhat hard to follow, especially around the proof sketches.
    2) The experimental section only accounts for synthetic data. Is it possible to add an experiment that includes real-world data? This will indeed highlight the applicabilities of your approach.

---

> ### Author Response · Authors · 2022-08-01
> **Review response**
>
> We would like to thank the reviewer for their effort and for their constructive comments. Below we provide answers to specific questions.
>
> **Q:** However, a valid question that usually pops up among practitioners who aim to use dimensionality reduction to boost the performance of their work, are matrices with linear decay/exponential decay common in the field of data science, i.e., how often we get to actually deal with such data that admits such properties?
>
> **A:**
> This is a good question regarding the general applicability of methods using such techniques (including Hutch++ [24], the  diagonal estimator of [5], the algorithms of [26]). An example application where the matrices inherently have large decay is related to graphs, specifically for computing node centralities/importances, as they relate to the exponential of the adjacency matrix $e^A$. Approximations of the matrix exponential in the matrix-vector query model also exist, e.g. via polynomial expansions or Lanczos methods for function approximations. For further examples we can also directly point to the related Hutch++ paper of Meyer et al [24], where they do present experiments on real-world matrices with spectral decay. Their algorithms, however, solve a different problem, i.e. they compute matrix traces. At the time of this writing, our work on real data is still in a preliminary stage (it also involves quite some theoretical analysis for the applications that we are working on).
>
>
> **Q:**
> In addition, is it possible to use some variant of your analysis when dealing with lewis weights?
>
> **A:**
> This is a very interesting point that we have not thought of. We are not entirely sure what the question refers to, is it about approximating the Lewis weights? If yes, note that for $p=2$, Lewis weights are equal to the leverage scores, which are detailed in Section 4. For general $p$, it is unclear for us at this point, but it is definitely an interesting question for future research.
>
> **Q:**
> Finally, please make sure that the main manuscript is at most 9 pages, while the current version of the paper can be refered to as the supplementary material.
>
> **A:**
> Regarding this, we initially followed the official guidelines, from which it appears that adding an Appendix to the main document is allowed. Quoting the official website:
>
> "Can I include appendices in my main submission file? Yes. You can include appendices with the main submission file, or you can include them as a separate file in the supplementary materials."
> (cf. https://nips.cc/Conferences/2022/PaperInformation/NeurIPS-FAQ).
>
> However, if this is not the case, we are willing to make all the appropriate changes as needed.
>
> **Conclusion:**
> We hope that we have answered all of the reviewer's questions as best as possible. We want to thank them again for their detailed comments and their constructive criticism.

---

> > ### Author Response · Authors · 2022-08-08
> > **Further Questions**
> >
> > As the deadline for the Reviewer-Author discussion phase approaches, we were wondering if our responses were sufficient or if there are any further clarifications needed. Let us know, we are happy to answer any further questions.

---

> > > ### Comment · Reviewer_Lczq · 2022-08-08
> > > **Satisfactory responses**
> > >
> > > I thank the authors for their feedback. I have decided to increase my score in light of their responses.

---

> > > > ### Author Response · Authors · 2022-08-09
> > > > **Thank you - Rebuttal Acknowledgement**
> > > >
> > > > Thank you for taking our responses into consideration and for raising the score! Just a note, there appears to be this "Rebuttal Acknowledgement" button for the reviewers, we are not sure if it's mandatory or if it has any meaning in general, but we thought to mention just in case. Thanks again

---

### Official Review · Reviewer_GYrC · 2022-07-12

**Rating:** 5
**Confidence:** 3
**Soundness:** 4 excellent
**Presentation:** 3 good
**Contribution:** 3 good

**Summary:**

The proposed problem is to estimate the euclidean length of matrix rows, and the authors propose a solution using fewer queries than requires by the JL-transform. This is done in the "matrix-vector query model", where access to the target matrix A is indirect, and only via queries of the form Ax for any vector x. The authors suggest an algorithm based on the Hutch++ algorithm, and give some claims of optimality. Results are derived for estimating Frobenius norms, inter-row distances, and statistical leverage scores.

**Questions:**

What is the main algorithmic contribution above the Hutch++ algorithm?

Can you further elaborate on the import of Lemma 3, and what it means for the optimality of Theorem 2?

--------------------------------

Following the author rebuttal, and explanation on contribution above the Hutch++ algorithm, I've raised my score to 5.

Concerning the statement of the JL lemma: Your Definition 1 relates to the norm of the input vectors. JL relates to the distance between the input vectors. This isn't the same, and the latter can be maintained without the former.

**Strengths And Weaknesses:**

Strengths: Rigorous approach to several interesting mathematical problems.

Weaknesses: The authors fail to argue that this paper has sufficient (in fact, any) ties to other machine learning applications. In my opinion the paper is out of scope for this conference, and seems more suited to a cs theory conference. The matrix-vector sample model is also quite limited, and in general I'm also unsure of how much this is a contribution this paper makes over those of the Hutch++ algorithm. The results seem incremental.

Minor concern: The statement of the JL lemms is not precise, as the lemma itself relates to distances and the statement relates to norms. It is true that norms imply distances, but only if the transform is linear, and only if one takes the input set to be all difference vectors, which is a larger set.

---

> ### Author Response · Authors · 2022-08-01
> **Review response**
>
> We would like to thank the reviewer for their effort and for the detailed comments. Below we provide detailed clarifications for the questions and concerns raised.
>
> **Q:** What is the main algorithmic contribution above the Hutch++ algorithm?
>
> **A:**
> We will start with this question as it appears to be the primary concern, and it should also highlight the novelty of this work. Hutch++, as well as Hutchinson's original algorithm, are estimators for the trace of a matrix. Matrix trace estimation, albeit related, is a different problem than the problems discussed in this paper. We present algorithms for estimating (i) Euclidean row norms (ii) Euclidean distances and (iii) leverage scores. Several techniques that we use are inspired by those of Hutch++, and this is clearly highlighted in the text, but it addresses different problems.
> A more intuitive example between the connections and differences between our work and Hutch++ would be the following:
>
> - Hutch++, as a trace estimator, can be used to estimate the trace of $AA^\top$, which is equal to the squared Frobenius norm of $A$. But it can only estimate the trace, which is the sum of the diagonal elements (or, equivalently, the sum of the eigenvalues). It cannot estimate, however, individual diagonal elements, individual eigenvalues, etc.
>
> - Our work, on the other hand, can be used to estimate the squared Euclidean row norms of $A$, which are equal to the diagonal elements of $A A^\top$. Summing up those individual estimations, we can also get an estimation of the trace of $AA^\top$, which is what Hutch++ returns, if it is applied on $AA^\top$.
> We hope that this clarifies the reviewer's concern.
>
> **Q:** Can you further elaborate on the import of Lemma 3, and what it means for the optimality of Theorem 2?
>
> **A:**
> This is a very interesting question. First we need to clarify that we do not claim that Theorem 2 is optimal. Lemma 3 (as well as Section II "Limitations of low-rank projections" in the Appendix) provide some evidence towards that direction, but it is not a formal proof of optimality, it is more of a conjecture. More specifically, they demonstrate that if someone uses low-rank projections, like we do, then there exist corner cases that make it highly unlikely for any such algorithm to do any better.
> Moreover, standard JL approximations have been proven to be optimal for the worst-case (element-wise), and  Hutch++ is also optimal for trace estimation. All of these hint that this work should be at least nearly-optimal, but at this point we do not have all the required lower bounds to make a concrete optimality argument. It is definitely a very interesting question for future work.
>
> **Q:** The authors fail to argue that this paper has sufficient (in fact, any) ties to other machine learning applications. In my opinion the paper is out of scope for this conference, and seems more suited to a cs theory conference.
>
> **A:**
> This might not be entirely true, we do mention four very successful publications in the introduction which are direct applications of random projections in ML problems, e.g. $k$-means clustering; cf. [2, 6, 16, 11]. Three of those are in ML conferences/journals: NeurIPS, ICML, Machine Learning. There exist many more publications using similar techniques in ML conferences/journals. If we have missed some important references, we would definitely be willing to add them. In general, we do believe that dimensionality reduction is highly relevant to the conference, and that our work is well within context.
>
> **Q:** The statement of the JL lemms is not precise, as the lemma itself relates to distances and the statement relates to norms. It is true that norms imply distances, but only if the transform is linear, and only if one takes the input set to be all difference vectors, which is a larger set.
>
> **A:** We are not sure which exact statement this question refers to, we have reviewed all the statements in the paper and we hope that they are all clear and precise. In general, we have tried to make it as clear as possible in the article how we distinct between estimation of row norms and estimation of Euclidean distances. We have split it in two different sections, each one with its own proofs and analysis. We can also point out that Dasgupta and Gupta (see ref. [R1]) use vector lengths as a building block in, what we believe to be, the most popular proof of the JL Lemma.
>
> **Conclusion:**
> We hope that we were able to answer all of the reviewer's questions. We would like to thank them again for their constructive comments.

---

> > ### Author Response · Authors · 2022-08-01
> > **References for review response**
> >
> > **References:**
> >
> > [R1] Dasgupta, Sanjoy, and Anupam Gupta. "An elementary proof of a theorem of Johnson and Lindenstrauss." Random Structures \& Algorithms 22.1 (2003): 60-65.

---

> > > ### Author Response · Authors · 2022-08-08
> > > **Further questions**
> > >
> > > As the deadline for the Reviewer-Author discussion phase approaches, we were wondering if our responses were sufficient or if there are any further clarifications needed. Let us know, we are happy to answer any further questions.

---

### Meta-Review · Area_Chair_95Sc · 2022-08-26

**Recommendation:** Accept
**Confidence:** Certain

**Metareview:**

All reviews for this paper were positive, albeit with a varying level of enthusiasm. Reviewers found the problem, the results (both theoretical and experimental) and the techniques (very) interesting. The main concerns were whether the paper is a good fit for the conference (given that dimensionality reduction is more of a machine learning tool rather a machine learning problem per se) and lack of experiments on real data. But ultimately, the positives significantly outweighed the negatives.

**Award:**

No

---

### Decision · Program_Chairs · 2022-09-14

Accept